# Credibility-Aware Weighting Federated Causal Discovery for Time Series

**Jiegang Xu**[1]   **Fuyuan Cao**[1]   **Jiye Liang**[2]

## Abstract

Federated causal discovery for time series is becoming increasingly important in many application domains. In practice, intervention policies on each client often change over time, causing the local underlying causal mechanisms to drift rather than remain fixed. Moreover, different sampling frequencies across clients yield incompatible time scales in the observed data, making the resulting local causal graphs naturally heterogeneous and difficult to aggregate consistently. Accordingly, we propose Fed-CAW, a Credibility-Aware Weighting Federated causal discovery framework for time series. Specifically, we define edge-level credibility scores that quantify per-edge reliability by summarizing (i) within-client temporal stability across windows and (ii) cross-client temporal consistency after mapping heterogeneous sampling frequencies onto a unified time scale. We then aggregate privatized edge statistics under differential privacy, treating credibility scores as weights to recover a global causal graph while preserving personalized local structures without sharing raw data. Theoretically, we demonstrate the rationale for the unified time scale mapping and establish rigorous differential privacy guarantees. Experimental results on synthetic and real-world datasets demonstrate the effectiveness of our proposed method.

## 1. Introduction

Causal discovery from time series plays a significant role in various disciplines and industrial fields, including biology, epidemiology, and information-physical systems (Duan et al., 2023; Absar et al., 2023; Castri et al., 2024). Understanding how variables interact across multiple temporal scales is crucial for tasks such as treatment planning, risk management, and policy evaluation (Wu et al., 2024; Varambally et al., 2024; Jin et al., 2024). For example, medication administration and vital signs over the past few hours jointly determine the current blood pressure; interest rate movements over the previous days affect stock returns at different temporal scales.

In modern distributed applications, time series data are rarely consolidated in a single data center. Instead, the data are typically siloed across multiple clients, which are unwilling or even legally prohibited from sharing raw records due to privacy, regulatory, or commercial considerations (Saha et al., 2022; Jia et al., 2022; Cyffers et al., 2024). In this context, federated causal structure learning (CSL) has attracted increasing attention as a framework for recovering a global causal structure from distributed observations without centralizing the raw data. Most existing federated CSL methods follow a common paradigm: they adapt classical static approaches, including constraint-based, score-based, and continuous-optimization methods, to the multi-client setting and aggregate local graphs, scores, or gradients under communication and privacy constraints (Ng & Zhang, 2022; Mian et al., 2023; Gao et al., 2023; Chen et al., 2026). Despite the remarkable progress, most existing federated causal discovery methods are built on two inherent assumptions: the underlying causal mechanisms on each client are essentially time invariant, and the observations across clients are collected at a common sampling frequency.

However, in practical federated applications, local intervention policies often exhibit substantial differences across clients and are typically adjusted over time (Yang et al., 2022; Zheng et al., 2024; Ferdous et al., 2025). The evolution can make the relationships among observed variables vary across periods, which in turn interferes with reliable identification of the overall causal structure for each client. For instance, a limited time discount campaign in a supermarket can simultaneously increase the observed correlation among sales of cola, chips, and other products with similar demand patterns; likewise, a temporary shift to a more aggressive medication regimen for a subset of patients at high risk can markedly strengthen the dependence between clinical indicators such as blood pressure and heart rate during

---

[1]School of Computer and Information Technology, Shanxi University, Taiyuan, China [2]Key Laboratory of Computational Intelligence and Chinese Information Processing of Ministry of Education, School of Computer and Information Technology, Shanxi University, Taiyuan, China. Correspondence to: Fuyuan Cao <cfy@sxu.edu.cn>.

*Proceedings of the 43rd International Conference on Machine Learning*, Seoul, South Korea. PMLR 306, 2026. Copyright 2026 by the author(s).

that phase. Such brief hot spot periods amplify local statistical associations; if temporal constraints and robustness calibration across windows are absent, these temporary phenomena can easily be mistaken for long-term stable causal relationships and subsequently become embedded in the global causal graph through federated aggregation.

In addition, heterogeneous sampling frequencies further complicate aggregation across clients: the same underlying causal influence may be observed at very different discrete time resolutions, making the lag structures estimated on different clients difficult to align (Gong et al., 2015; Tank et al., 2019; Cheng et al., 2024). For instance, tertiary hospitals typically monitor vital signs at high frequency, whereas community hospitals or primary care clinics record the same indicators much less frequently; as a result, the same disease trajectory admits quite different discrete-time representations across clients. Directly aggregating results obtained under such disparate sampling intervals can mischaracterize both the timing and the strength of causal effects, thereby undermining reliable recovery of the global causal structure.

In this work, we propose Fed-CAW, which combines temporally smoothed local structure learning with credibility-aware aggregation under differential privacy to recover a global causal graph and a personalized local graph for each client without sharing raw data. On the client side, we fit structural models with directed acyclic constraints on sliding windows and impose smoothness penalties across adjacent windows to obtain stable local causal graphs, and perform continuous-time reparameterization based on matrix logarithms of lag augmented operators to map edges estimated under different sampling intervals onto a common reference time scale. Based on these representations, we define edge-level credibility scores that quantify the reliability of each estimated edge by summarizing (i) *within-client temporal stability* across windows and (ii) *cross-client temporal consistency* of its lag profile after mapping heterogeneous sampling intervals to the unified reference time scale. On the server side, we aggregate privatized edge-level statistics using credibility based weights to recover a global causal graph and estimate per edge cross-client sharing probability to distinguish broadly shared global relations from client-specific effects, which are fed back to guide subsequent personalized updates. Extensive experiments on synthetic and real-world datasets validate the effectiveness of Fed-CAW.

## 2. Formal Preliminaries

Consider a set of realizations of a multivariate time series, with each individual realization of size $T$ in the form of $\mathbf{X}_t := [x_{t,i}]_{i=1}^d \in \mathbb{R}^d$. Here $t \in \{0, 1, \ldots, T\}$ represents the time index, $\mathbf{X}_t$ represents the observed values of all $d$ variables at time $t$ in an observational time series dataset.

**Federated causal structure learning setting.** We consider a federated environment with $M$ clients $\mathcal{C} = \{\mathcal{C}_1, \mathcal{C}_2, \ldots, \mathcal{C}_M\}$ and one central server $S$. Client $\mathcal{C}_m$ holds a local dataset $D^{(m)}$ consisting of observations on a common set of $d$ variables $V = \{1, 2, \ldots, d\}$, but raw data never leave the client. Let $n_m$ denote the number of effective samples on client $\mathcal{C}_m$. Following the standard federated CSL formulation, the union of local variables coincides with $V$ and the goal is to learn a global causal graph $\mathcal{G}^{\text{glob}}$ over $V$ from all $\{D^{(m)}\}_{m=1}^M$ without centralizing the data, by exchanging only intermediate statistics or model updates with the server instead of raw samples.

In this work, we focus on the common "horizontal" setting where all clients share the same variable set $V$ but may have different sample sizes, sampling frequencies and data distributions.

**Causal structure learning for multivariate time series.** For each client $\mathcal{C}_m$, we observe a multivariate time series $\mathbf{X}_t^{(m)}$, $m \in \{1, 2, \ldots, M\}$, where $t \in \{0, 1, \ldots, T_m\}$ represents the time index of the observation by client $\mathcal{C}_m$. We characterize $\mathbf{X}^{(m)}$ by a standard structural vector autoregressive model (Pamfil et al., 2020; Sun et al., 2023):

$$\mathbf{X}_t^{(m)} = \mathbf{X}_t^{(m)}\mathbf{W}^{(m)} + \sum_{\ell=1}^L \mathbf{X}_{t-\ell}^{(m)}\mathbf{A}_\ell^{(m)} + \mathbf{Z}_t^{(m)}, \quad (1)$$

where $L$ is the autoregressive order (maximum lag). The term $\mathbf{Z}_t^{(m)} \in \mathbb{R}^d$ is a $d$-dimensional noise vector drawn from a continuous distribution. We assume that $\mathbf{Z}_t^{(m)}$ is independent of $\mathbf{Z}_{t'}^{(m)}$ for $t' \neq t$ and of $\mathbf{X}_{t'}^{(m)}$ for all $t' < t$. The matrices $\mathbf{W}^{(m)}$ and $\mathbf{A}_\ell^{(m)} \in \mathbb{R}^{d \times d}$ represent weighted adjacency matrices for the intra-slice and inter-slice edges in $\mathcal{G}$, respectively. We follow the continuous constrained optimization reformulation of DAG learning in (Chen et al., 2026), which for each client $\mathcal{C}_m$ leads to the following problem with an explicit acyclicity constraint:

$$\min_{\mathbf{W}^{(m)}, \{\mathbf{A}_\ell^{(m)}\}} \mathcal{L}_{\text{data}}^{(m)}\big(\mathbf{W}^{(m)}, \{\mathbf{A}_\ell^{(m)}\}_{\ell=1}^L\big) + \lambda_W \big\|\mathbf{W}^{(m)}\big\|_1$$

$$+ \lambda_A \sum_{\ell=1}^L \big\|\mathbf{A}_\ell^{(m)}\big\|_1 \quad \text{s.t.} \quad h\big(\mathbf{W}^{(m)}\big) = 0.$$

where $\mathcal{L}_{\text{data}}^{(m)}$ is typically a squared-error or negative log-likelihood loss induced by Eq. (1), $\|\cdot\|_1$ denotes the elementwise $\ell_1$-norm that promotes sparsity, and

$$h\big(\mathbf{W}^{(m)}\big) = \text{tr}\Big(e^{\mathbf{W}^{(m)} \circ \mathbf{W}^{(m)}}\Big) - d,$$

is a smooth acyclicity function that enforces the learned network to be acyclic, where $\circ$ denotes the Hadamard product. The zero level set of $h(\cdot)$ characterizes weighted DAGs.

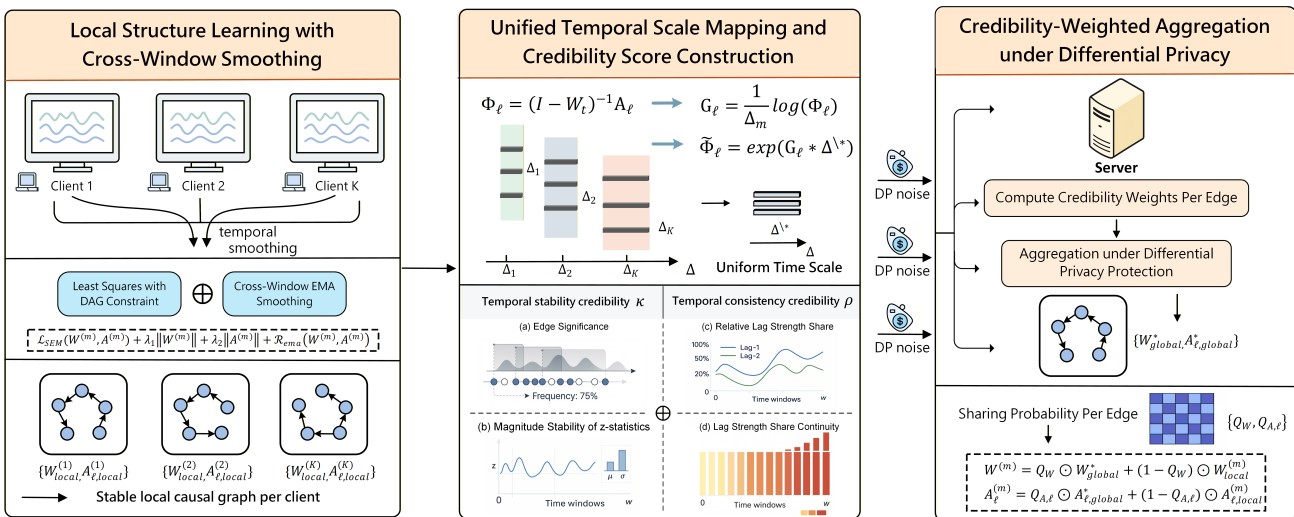

*Figure 1.* Overview of the proposed Fed-CAW framework. Left: each client fits a temporally smoothed local SVAR with cross-window temporal regularization to obtain a stable local causal graph. Middle: window-wise parameters are mapped to a unified temporal scale and used to construct edge-level effect statistics together with temporal stability and temporal consistency credibility scores $(\kappa, \rho)$. Right: privatized edge statistics are aggregated on the server through credibility-weighted meta analysis under differential privacy, yielding a global backbone graph and edge-level sharing probabilities that softly fuse global and local structures for personalization.

## 3. The Proposed Fed-CAW Algorithm

This section presents Fed-CAW, which aims to learn a relatively stable local causal structure on each client and further recover a global causal graph under temporal consistency. The detailed algorithm for the full procedure is provided in Appendix C.

### 3.1. Learning Objective

In this paper, we consider a setting in which the underlying causal mechanisms within each client may drift over time due to short-term interventions and other forms of non-stationarity. Rather than recovering a separate graph at every time point, we aim to extract, for each client, a local **backbone** causal structure whose edges persist over a substantial portion of the observation period. These client-specific backbones are then aggregated across clients in a federated manner to yield a global causal graph.

Formally, each client $\mathcal{C}_m$ holds a multivariate time series $\mathbf{X}_t^{(m)}$ of length $T_m$ over the common variable set $V = \{1, 2, \ldots, d\}$ that, over approximately stable time intervals, follows the structural equation model (SEM) in Eq. (1). Our goal is to jointly recover:

- a set of *local* causal graphs $\{\mathcal{G}_{\text{local}}^{(m)}\}_{m=1}^M$ over $V$, allowing personalized deviations from $\mathcal{G}^{\text{glob}}$.

- a *global* causal graph $\mathcal{G}^{\text{glob}}$ over $V$, capturing edges that are consistently supported by multiple clients;

without sharing raw sequences across clients.

### 3.2. Time-smoothed Local Structure Learning

We first describe the local learning module on each client, which produces window-wise SEM estimates and edge-level summaries without revealing raw data.

**Sliding-window SEM.** For each client $\mathcal{C}_m$, we choose a shared window length $G$ and stride $s$ to construct local temporal windows. Specifically, the $k$-th window is defined as

$$\mathcal{W}_k^{(m)} = \{t_k, \ldots, t_k + G - 1\}, \quad t_k = L + 1 + (k-1)s,$$

where $k = 1, \ldots, K_m$. Here, $G$ determines the number of observations used for each local structure estimate, while $s$ specifies the spacing between the starting points of two consecutive windows. Within the $k$-th window, the observations are represented by a local linear SVAR of the form

$$\mathbf{X}_t^{(m)} = \mathbf{X}_t^{(m)}\mathbf{W}_k^{(m)} + \sum_{\ell=1}^{L} \mathbf{X}_{t-\ell}^{(m)}\mathbf{A}_{k,\ell}^{(m)} + \mathbf{Z}_{k,t}^{(m)}, \quad t \in \mathcal{W}_k^{(m)}.$$

(2)

Here, $\mathbf{W}_k^{(m)}$ and $\mathbf{A}_{k,\ell}^{(m)}$ denote the weighted adjacency matrices of the intra-slice and inter-slice edges for client $\mathcal{C}_m$ within window $\mathcal{W}_k^{(m)}$, respectively.

**Cross-window temporal smoothing.** A naive window-wise application of Eq. (2) may overfit short-term "hotspots" and local noise bursts, resulting in highly unstable structures. To stabilize local graphs while still allowing gradual structural drift, we introduce smoothness penalties that couple adjacent windows. Specifically, for each client $\mathcal{C}_m$ we optimize

the following temporally smoothed objective:

$$
\begin{aligned}
\min_{\{\mathbf{W}_k^{(m)}, \mathbf{A}_{k,\ell}^{(m)}\}} & \sum_{k=1}^{K_m} \Big[ \mathcal{L}_{\text{data}}^{(m,k)}\big(\mathbf{W}_k^{(m)}, \{\mathbf{A}_{k,\ell}^{(m)}\}\big) + \lambda_W \big\|\mathbf{W}_k^{(m)}\big\|_1 \\
& + \lambda_A \sum_{\ell=1}^{L} \big\|\mathbf{A}_{k,\ell}^{(m)}\big\|_1 \Big] + \gamma_W \sum_{k=2}^{K_m} \big\|\mathbf{W}_k^{(m)} - \mathbf{W}_{k-1}^{(m)}\big\|_F^2 \\
& + \gamma_A \sum_{\ell=1}^{L} \sum_{k=2}^{K_m} \big\|\mathbf{A}_{k,\ell}^{(m)} - \mathbf{A}_{k-1,\ell}^{(m)}\big\|_F^2 \\
\text{s.t.} \quad & h\big(\mathbf{W}_k^{(m)}\big) = 0, \quad k = 1, \dots, K_m,
\end{aligned}
\tag{3}
$$

where $\mathcal{L}_{\text{data}}^{(m,k)}$ denotes a squared-error or negative log-likelihood loss on window $\mathcal{W}_k^{(m)}$, $\|\cdot\|_F$ is the Frobenius norm, and $h(\cdot)$ is the smooth acyclicity function.

The smoothing terms controlled by $(\gamma_W, \gamma_A)$ encourage successive windows to share similar structures while still allowing slow drift. In practice, Eq. (3) can be optimized via stochastic gradient methods, and combined with a penalty-based treatment of the acyclicity constraints in each window.

**Local edge-level statistics aggregation.** The smoothed window-wise estimates $\{\hat{\mathbf{W}}_k^{(m)}, \hat{\mathbf{A}}_{k,\ell}^{(m)}\}_{k=1}^{K_m}$ provide candidate conditioning sets for testing directed interactions. For each candidate edge $e$, we perform Gaussian conditional-independence tests and obtain window-wise Fisher $z$ statistics $\{z_{k,e}^{(m)}\}_{k=1}^{K_m}$, which offer scale normalized edge evidence and compact summaries for server aggregation and remain compatible with the subsequent continuous-time alignment and credibility weighting.

To obtain a relatively stable structure at the client level, we aggregate the window-wise evidence via a weighted average

$$
z_e^{(m)} = \sum_{k=1}^{K_m} \omega_{k,e}^{(m)} z_{k,e}^{(m)}, \qquad \sum_{k=1}^{K_m} \omega_{k,e}^{(m)} = 1,
\tag{4}
$$

where $\omega_{k,e}^{(m)} \geq 0$ (e.g., uniform or proportional to effective window sample sizes). Subsequently, we perform significance testing and obtain the client-specific causal graph $\mathcal{G}_{\text{local}}^{(m)} = (V, \mathcal{E}_{\text{loc}}^{(m)})$.

While $z_e^{(m)}$ summarizes average evidence, stability also depends on how consistently the evidence is sustained over time. For each $e \in \mathcal{E}_{\text{loc}}^{(m)}$, we consider the support frequency across windows and the stability of evidence magnitudes. Given a relevance threshold $\tau_{\text{loc}} > 0$, define

$$
\mathcal{S}_e^{(m)} = \Big\{ k \in \{1, \dots, K_m\} : |z_{k,e}^{(m)}| > \tau_{\text{loc}} \Big\},
\tag{5}
$$

and the Jeffreys smoothed support frequency

$$
\text{Fre}_e^{(m)} = \frac{|\mathcal{S}_e^{(m)}| + \frac{1}{2}}{K_m + 1}.
\tag{6}
$$

Magnitude stability summarizes the temporal variation of

$$
\text{Stab}_e^{(m)} = \left[ 1 + \frac{\text{MAD}\big(\{|z_{k,e}^{(m)}|\}_{k \in \mathcal{S}_e^{(m)}}\big)}{\text{median}\big(\{|z_{k,e}^{(m)}|\}_{k \in \mathcal{S}_e^{(m)}}\big)} \right]^{-1},
\tag{7}
$$

with $\text{Stab}_e^{(m)} := 0$ when $|\mathcal{S}_e^{(m)}| < 2$.

**Definition 3.1** (Temporal stability credibility). For client $\mathcal{C}_m$ and edge $e \in \mathcal{E}_{\text{loc}}^{(m)}$, define its within client stability credibility as the geometric mean

$$
\kappa_e^{(m)} = \sqrt{\text{Fre}_e^{(m)} \, \text{Stab}_e^{(m)}} \in [0, 1].
\tag{8}
$$

Here, $\text{Fre}_e^{(m)}$ represents support frequency across windows and $\text{Stab}_e^{(m)}$ represents temporal stability of evidence magnitudes. Consequently, $\kappa_e^{(m)}$ has high credibility only when both frequency and stability are high.

### 3.3. Continuous-time Reparameterization and Edge Credibility

Clients may operate at different sampling intervals, so window parameters estimated at native time steps lie on different time scales across clients. Accordingly, we rescale each window to a shared reference step $\Delta_{\text{ref}}$ via continuous-time reparameterization, and then extract edge-level credibility signals on the aligned scale.

**Continuous-time operator alignment.** For each client $\mathcal{C}_m$ and window $\mathcal{W}_k^{(m)}$, we construct a lag augmented linear operator $\Phi_k^{(m)}$ from the fitted windowed time series SEM in Eq. (2). Specifically, $\Phi_k^{(m)} \in \mathbb{R}^{d(L+1) \times d(L+1)}$ takes the companion form

$$
\Phi_k^{(m)} = \begin{bmatrix} W_k^{(m)\top} & A_{k,1}^{(m)\top} & \cdots & A_{k,L}^{(m)\top} \\ I_d & 0 & \cdots & 0 \\ 0 & I_d & \cdots & 0 \\ \vdots & & \ddots & \vdots \\ 0 & 0 & \cdots & I_d \; 0 \end{bmatrix},
\tag{9}
$$

where $I_d$ denotes the $d \times d$ identity matrix. Let $\Delta_m$ denote the sampling interval of client $\mathcal{C}_m$. Within window $\mathcal{W}_k^{(m)}$, we assume that the augmented state dynamics admit a continuous-time generator $G_k^{(m)}$ such that the discrete operator satisfies

$$
\Phi_k^{(m)} \approx \exp\Big(\Delta_m G_k^{(m)}\Big).
\tag{10}
$$

When $\Phi_k^{(m)}$ is stable and the principal matrix logarithm exists, we estimate

$$
\hat{G}_k^{(m)} := \Delta_m^{-1} \log\Big(\Phi_k^{(m)}\Big),
\tag{11}
$$

and map the operator to the shared reference step $\Delta_{\mathrm{ref}}$ via

$$\widetilde{\Phi}_k^{(m)} = \exp\left(\Delta_{\mathrm{ref}} \hat{G}_k^{(m)}\right) \approx \exp\left(\frac{\Delta_{\mathrm{ref}}}{\Delta_m} \log\left(\Phi_k^{(m)}\right)\right).$$

Next, we extract the first block row of $\widetilde{\Phi}_k^{(m)}$ and denote its $d \times d$ blocks by $\{\widetilde{\Phi}_{k,\ell}^{(m)}\}_{\ell=0}^L$, where $\ell = 0$ corresponds to contemporaneous effects and $\ell \geq 1$ to lag $\ell$ effects on the common time scale $\Delta_{\mathrm{ref}}$. If short windows or high lag orders lead to high estimation variance, we optionally fit an exponential decay curve to smooth stability related indicators (Appendix B.3), without changing the edge statistics sent to the server.

**Lag profile consistency on a unified time scale.** The aligned coefficient blocks $\{\widetilde{\Phi}_{k,\ell}^{(m)}\}_{\ell=0}^L$ provide a lag resolved description of each directed interaction. For edge $e = (i \to j)$ in window $\mathcal{W}_k^{(m)}$, we form the lag profile magnitude vector

$$\mathbf{a}_{k,e}^{(m)} = \left[|\widetilde{\Phi}_{k,0,ij}^{(m)}|, |\widetilde{\Phi}_{k,1,ij}^{(m)}|, \ldots, |\widetilde{\Phi}_{k,L,ij}^{(m)}|\right]^\top \in \mathbb{R}_+^{L+1}, \tag{12}$$

and normalize it as

$$\boldsymbol{\pi}_{k,e}^{(m)} = \frac{\mathbf{a}_{k,e}^{(m)}}{\mathbf{1}^\top \mathbf{a}_{k,e}^{(m)} + \epsilon}, \tag{13}$$

where $\epsilon > 0$ is a small constant for numerical stability. Averaging over the selected windows $\mathcal{S}_e^{(m)}$ in Eq. (5) yields

$$\bar{\boldsymbol{\pi}}_e^{(m)} = \frac{1}{|\mathcal{S}_e^{(m)}|} \sum_{k \in \mathcal{S}_e^{(m)}} \boldsymbol{\pi}_{k,e}^{(m)}. \tag{14}$$

**Definition 3.2** (Temporal consistency credibility). For edge $e \in \mathcal{E}_{\mathrm{loc}}^{(m)}$, define its lag profile consistency on the unified time scale as

$$\rho_e^{(m)} = \frac{1}{|\mathcal{S}_e^{(m)}|} \sum_{k \in \mathcal{S}_e^{(m)}} \frac{\langle \boldsymbol{\pi}_{k,e}^{(m)}, \bar{\boldsymbol{\pi}}_e^{(m)} \rangle}{\|\boldsymbol{\pi}_{k,e}^{(m)}\|_2 \|\bar{\boldsymbol{\pi}}_e^{(m)}\|_2} \in [0, 1]. \tag{15}$$

A larger $\rho_e^{(m)}$ indicates a more consistent lag allocation across windows on the unified time scale. The server uses $\rho_e^{(m)}$ together with the within client stability credibility $\kappa_e^{(m)}$ to modulate aggregation weights.

**Note.** We emphasize that the continuous-time reparameterization is applied independently to each client: client $\mathcal{C}_m$ is associated with its own generator $G_k^{(m)}$ in each window. The alignment step maps these client-specific generators to the shared reference step $\Delta_{\mathrm{ref}}$, but does not require the generators, or the resulting discrete-time graphs, to be identical across clients.

## 3.4. Credibility-weighted Aggregation under Differential Privacy

This section describes server aggregation of edge evidence. Each client transmits compact edge statistics. The server fuses these statistics through credibility weights under differential privacy.

**Private edge signatures.** For each edge $e$, client $\mathcal{C}_m$ forms a signature $s_e^{(m)} = (z_e^{(m)}, \kappa_e^{(m)}, \rho_e^{(m)})$. We apply $(\varepsilon, \delta)-$ differential privacy to the evidence transmitted by clients to satisfy privacy constraints.

$$\widetilde{z}_e^{(m)} = z_e^{(m)} + \xi_{z,e}^{(m)}, \qquad \xi_{z,e}^{(m)} \sim \mathcal{N}(0, \sigma_z^2), \tag{16}$$

where $\xi_{z,e}^{(m)}$ are mutually independent across edges. The variance $\sigma_z^2$ is calibrated via the Gaussian mechanism (Theorem 4.5) according to the global sensitivity bound in Proposition 4.3.

**Definition 3.3** (Credibility-weighted server aggregation). For each edge $e$, the server assigns a credibility weight to client $\mathcal{C}_m$ as

$$w_e^{(m)} = \frac{n_m}{\sum_{r=1}^M n_r} \kappa_e^{(m)} \rho_e^{(m)}, \qquad \alpha_e^{(m)} = \frac{w_e^{(m)}}{\sum_{r=1}^M w_e^{(r)}},$$

and aggregates the noisy evidence via

$$z_e^{\mathrm{meta}} = \sum_{m=1}^M \alpha_e^{(m)} \widetilde{z}_e^{(m)}. \tag{17}$$

The statistic $\widetilde{z}_e^{(m)}$ places evidence on a comparable scale under privacy. The credibilities $\kappa_e^{(m)}$ and $\rho_e^{(m)}$ emphasize contributions that remain stable across windows and consistent on the aligned time scale. A global significance test on $z_e^{\mathrm{meta}}$ then yields the global graph $\mathcal{G}^{\mathrm{glob}}$.

**Estimating cross-client sharing probabilities.** In federated time series applications, global and local causal graph structures need not coincide, so it is crucial to quantify how strongly each edge is shared across clients. We therefore define a sharing probability $Q_e \in [0, 1]$ that combines the breadth and directional agreement of the evidence:

$$Q_e = g\left(p_e^{\mathrm{share}}, C_e^{\mathrm{sign}}\right) \in [0, 1]. \tag{18}$$

Here, $p_e^{\mathrm{share}}$ reflects the effective fraction of clients supporting edge $e$, and $C_e^{\mathrm{sign}}$ measures the consistency of the evidence sign, $g(\cdot, \cdot)$ is instantiated as the geometric mean. Within Fed-CAW, $Q_e$ acts as a soft feedback signal, giving higher priority to edges with broad and coherent support than to those dominated by local effects.

## 4. Theoretical Analysis

In this section, we present a theoretical analysis of Fed-CAW, covering continuous-time alignment, the asymptotic

behavior of the credibility-weighted statistic, and the differential privacy guarantee for noisy edge evidence. All proofs and auxiliary lemmas are deferred to the appendix.

**Continuous-time alignment via matrix logarithms.** We first recall a basic property of matrix logarithms in Lemma B.4. Building on this lemma, the following result formalizes the alignment effect of Eq. (11) and shows that the recovered continuous-time generators are consistent within each client and comparable across clients.

**Proposition 4.1** (Continuous-time alignment via matrix logarithms). *Fix a window index $k$ and suppose that, for each client $\mathcal{C}_m$, the window-specific companion matrix $\Phi_k^{(m)}$ in Eq. (9) is stable, i.e., all eigenvalues lie strictly inside the unit disk. Then, by Lemma B.4, there exists a unique real matrix $G_k^{(m)}$ with eigenvalues having negative real parts such that*

$$\Phi_k^{(m)} = \exp\big(\Delta_m G_k^{(m)}\big), \qquad (19)$$

*where $\Delta_m > 0$ is the sampling interval of client $\mathcal{C}_m$. Let $\hat{G}_k^{(m)}$ and $\widetilde{\Phi}_k^{(m)}$ be the aligned generator and transition matrix defined in Section 3.3. Then the alignment step enjoys the following properties:*

*(i)* ***Within-client scale consistency.*** *If $\Phi_k^{(m)}$ is generated from $G_k^{(m)}$ via Eq. (19) and the principal matrix logarithm is used, then the alignment recovers the generator exactly and maps it to the common step size:*

$$\hat{G}_k^{(m)} = G_k^{(m)} \quad and \quad \widetilde{\Phi}_k^{(m)} = \exp\big(\Delta_{\mathrm{ref}} G_k^{(m)}\big).$$

*(ii)* ***Cross-client comparability.*** *Moreover, suppose that there exists a reference generator $G_k^\star$ such that*

$$\big\|G_k^{(m)} - G_k^\star\big\| \le \delta_k \qquad for\ all\ m, \qquad (20)$$

*and that all $G_k^{(m)}$ and $G_k^\star$ lie in a compact set on which the map $A \mapsto \exp(\Delta_{\mathrm{ref}} A)$ is Lipschitz continuous with constant $L_k$, i.e.,*

$$\big\|\exp(\Delta_{\mathrm{ref}} A) - \exp(\Delta_{\mathrm{ref}} B)\big\| \le L_k \|A - B\|.$$

*Then the aligned transition matrices satisfy*

$$\big\|\widetilde{\Phi}_k^{(m)} - \exp(\Delta_{\mathrm{ref}} G_k^\star)\big\| \le L_k\, \delta_k \qquad for\ all\ m,$$

*which shows that, after alignment, all clients remain within a controlled deviation of the same reference dynamics at the common time scale.*

**Asymptotic calibration of credibility-weighted testing.** After alignment, the server combines client-wise edge evidence using the credibility weights from Section 3.4. The next result shows that the resulting global test statistic is asymptotically standard normal under the global null, which supports significance testing.

**Theorem 4.2** (Asymptotic normality of the meta statistic). *For each edge $e$, let $z_e^{\mathrm{meta}}$ be the credibility-weighted meta statistic defined in Eq. (17), and its standardized form is given by*

$$Z_{meta} = \frac{z_e^{\mathrm{meta}}}{\sqrt{\big(1 + \sigma_z^2\big) \sum_{m=1}^M \big(\alpha_e^{(m)}\big)^2}}.$$

*Under the global null hypothesis that edge $e$ is absent for all clients, and under Assumption B.2, we have*

$$Z_{meta} \xrightarrow{d} \mathcal{N}(0,1) \qquad as \min_{m,k} n_{m,k} \to \infty.$$

*Moreover, under fixed nonzero local effects, $Z_{meta}$ has a nonzero limiting mean that increases with the aggregated signal strength and the breadth of cross-client support.*

**Sensitivity bounds and differential privacy.** The server-side fusion uses client-provided edge evidence. Fed-CAW perturbs this evidence before communication; the noise scale is set by bounding the sensitivity of the stacked evidence vector and then invoking the Gaussian mechanism.

**Proposition 4.3** (Bounded sensitivity of evidence vectors). *Under Assumption B.3, there exists a constant $B_z < \infty$ such that, for every client $\mathcal{C}_m$, window $k$, and edge $e$,*

$$\big|z_{k,e}^{(m)}\big| \le B_z.$$

*Consequently, the aggregated statistic $z_e^{(m)}$ in Eq. (4) also satisfies $|z_e^{(m)}| \le B_z$. For any pair of neighboring local datasets $D^{(m)}$ and $D^{(m)\prime}$ on client $\mathcal{C}_m$, let $z^{(m)}(D^{(m)})$ and $z^{(m)}(D^{(m)\prime})$ denote the corresponding stacked evidence vectors, formed by collecting all edge–level statistics as above. Then there exists a finite constant $\Delta_2$ such that*

$$\big\|z^{(m)}(D^{(m)}) - z^{(m)}(D^{(m)\prime})\big\|_2 \le \Delta_2.$$

**Definition 4.4** (Differential privacy for client-level evidence release). *A randomized mechanism $\mathcal{M}$ that maps a client's local dataset $D^{(m)}$ to an output in a space $\mathcal{O}$ satisfies differential privacy at the client level if, for any neighboring datasets $D^{(m)}, D^{(m)\prime}$ and any measurable subset $O \subseteq \mathcal{O}$,*

$$\Pr\big(\mathcal{M}(D^{(m)}) \in O\big) \le e^\varepsilon \Pr\big(\mathcal{M}(D^{(m)\prime}) \in O\big) + \delta.$$

**Theorem 4.5** (Differential privacy of Gaussian evidence mechanism). *Fix a client $\mathcal{C}_m$ and consider the mechanism that releases a noisy stacked evidence vector*

$$\mathcal{M}(D^{(m)}) = z^{(m)}(D^{(m)}) + \xi, \qquad \xi \sim \mathcal{N}(0, \sigma_z^2 I).$$

*Under Assumption B.3, Proposition 4.3 gives $\ell_2$-sensitivity at most $\Delta_2$. If*

$$\sigma_z \ge \frac{\Delta_2}{\varepsilon} \sqrt{2 \log \frac{1.25}{\delta}},$$

*then $\mathcal{M}$ satisfies client-level differential privacy.*

Taken together, Proposition 4.3 and Theorem 4.5 justify the noise calibration used in Eq. (16) and ensure that server-side aggregation relies only on privacy-preserving edge evidence.

# 5. Experiments

In this section, we empirically evaluate Fed-CAW on synthetic and real benchmark multi-client time series. Our synthetic studies rely on controlled simulations that allow us to vary (i) the amount of temporal drift and the strength and frequency of short-term interventions, and (ii) the heterogeneity in sampling rates across clients. We then report results on two real benchmarks, gene regulatory networks and fMRI connectivity, to assess whether the proposed method remains competitive in practical settings where privacy constraints prevent sharing raw data.

## 5.1. Experimental Setup

**Synthetic data generation.** We adopt a structural vector autoregressive (SVAR) model with time-varying parameters to generate client-specific non-stationary time series. For each client $\mathcal{C}_m$, the data are generated as

$$\mathbf{X}_t^{(m)} = \mathbf{X}_t^{(m)}\mathbf{W}_t^{(m)} + \sum_{\ell=1}^{L} \mathbf{X}_{t-\ell}^{(m)}\mathbf{A}_{\ell,t}^{(m)} + \mathbf{Z}_t^{(m)}. \quad (21)$$

We first construct a *backbone* intra-slice $\mathbf{W}_\star^{(m)}$ and inter-slice $\mathbf{A}_{\ell,\star}^{(m)}$ coefficient matrices, then generate time-varying parameters by superimposing short-term local interventions:

$$\mathbf{W}_t^{(m)} = \mathbf{W}_\star^{(m)} + \Delta\mathbf{W}_t^{(m)}, \quad (22)$$

$$\mathbf{A}_{\ell,t}^{(m)} = \mathbf{A}_{\ell,\star}^{(m)} + \Delta\mathbf{A}_{\ell,t}^{(m)}, \quad (23)$$

where $\Delta\mathbf{W}_t^{(m)}$ and $\Delta\mathbf{A}_{\ell,t}^{(m)}$ are sparse perturbations that are nonzero only on randomly selected intervention windows. These perturbations induce strong but transient changes in the conditional dependence structure within a client, mimicking local policy changes or temporary shocks. The backbone graphs $\mathbf{W}_\star^{(m)}$ and $\mathbf{A}_{\ell,\star}^{(m)}$ are sampled from directed Erdős–Rényi models (Erdős & Rényi, 1960) with a prescribed expected degree. We control the overall non-stationarity by varying the fraction of time under intervention and the magnitude of the perturbation entries. Concrete parameter choices are listed in Appendix D.

**Heterogeneous sampling.** To model heterogeneous sampling schemes, each client $\mathcal{C}_m$ is assigned a sampling interval $\Delta_m > 0$, and the continuous-time process underlying all clients is generated from a piecewise-stationary linear model with a common generator $G_\star$. The discrete-time SVAR parameters are then obtained via

$$\Phi^{(m)} = \exp(\Delta_m G_\star). \quad (24)$$

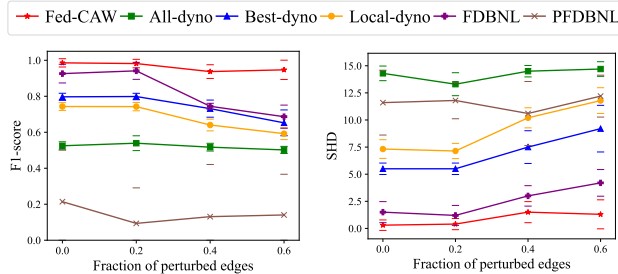

*Figure 2.* Experimental results on Synthetic I data.

and then map it to the $(W_\star^{(m)}, A_{\ell,\star}^{(m)})$ representation. During data generation, we vary the ratio between the largest and the smallest $\Delta_m$ to control the level of sampling heterogeneity. In heterogeneous sampling experiments, we explicitly misalign the client time grids and rely on the alignment based on the generator in Section 3.3 for aggregation.

**Baselines.** We compare Fed-CAW against both federated and centralized structure learning methods. On the federated side, we consider the FDBNL and PFDBNL (Chen et al., 2026). On the centralized side, we adopt methods based on Dynotears (Pamfil et al., 2020): Local-dyno, which fits a separate model on each client and reports performance by averaging client wise metrics; All-dyno, which pools all client data and fits a single centralized model; and best-dyno, an oracle that selects the client whose local model attains the smallest SHD to the ground truth.

**Evaluation metrics.** We report the Structural Hamming Distance (SHD) and F1-score, where lower SHD and higher F1-score indicate more accurate structure recovery. We also report AUROC and AUPRC as complementary metrics to assess overall causal discovery performance.

## 5.2. Datasets

We consider three synthetic datasets and two real-world datasets. Unless otherwise stated, we repeat each configuration $N_{\text{rep}} = 30$ times and report the mean SHD and F1-score with standard deviations. The corresponding AUROC and AUPRC results are provided in Appendix D.2.

**Synthetic I: Temporal drift under interventions.** We construct a suite with a fixed number of clients $M$ and variables $d$, and control nonstationarity by (i) the fraction of time steps affected by interventions and (ii) the magnitude of perturbations $(\Delta W_t^{(m)}, \Delta A_{\ell,t}^{(m)})$ added to the backbone parameters. This setting evaluates robustness to short-term yet potentially strong temporal shocks within clients.

**Synthetic II: Client scaling under limited observations.** We fix the total number of observations $\sum_m T_m$ and change the number of clients $M \in \{2, 4, 8, 16, 32, 64\}$, so that each client receives a shorter time series as $M$ increases.

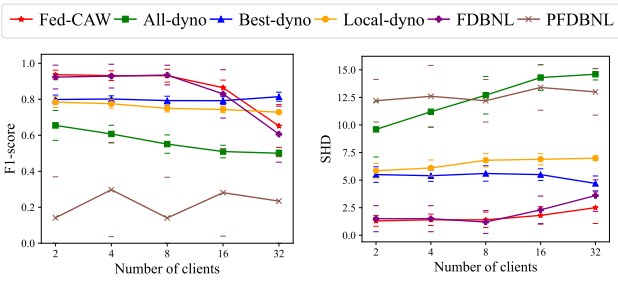

*Figure 3.* Experimental results on Synthetic II data.

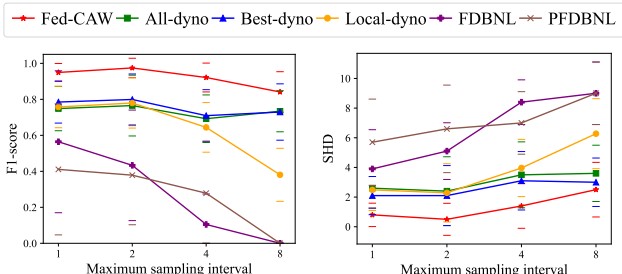

*Figure 4.* Experimental results on Synthetic III data.

This setting evaluates performance as local sample sizes shrink and local structure learning becomes increasingly underdetermined.

**Synthetic III: Global recovery with different sampling.** We generate data from a common continuous-time generator $G_\star$ and assign each client a sampling interval $\Delta_m \in [\Delta_{\min}, \Delta_{\max}]$ with heterogeneity ratio $\Delta_{\max}/\Delta_{\min} \in \{1, 2, 4, 8\}$. We then apply Eq. (24) to obtain time parameters, so that all clients share the same underlying dynamics on misaligned time grids. This suite evaluates how sampling heterogeneity affects global structure recovery and cross client aggregation.

**Functional Magnetic Resonance Imaging (fMRI).** The functional magnetic resonance imaging (Smith et al., 2011) forward model simulates realistic blood oxygen level dependent (BOLD) time series that arise from nonlinearly coupled causal dynamics. We consider simulations 1, 3, and 6 comprising $M = 50$ subjects treated as clients, with $d \in \{5, 10, 15\}$ regions of interest and sequence length $T \in \{200, 1200\}$. Subject level heterogeneity is introduced through subject specific interventions and mild variations in the underlying connectivity strengths.

**Dream3 Gene Network.** The Dream3 dataset (Prill et al., 2010) is a real systems biology benchmark that contains gene expression time series from five networks, including two E.coli networks and three Yeast networks, each with $D = 100$ genes. In this experiment, we split the time points of each experimental condition into five nonoverlapping segments and assign them to $M = 5$ clients to simulate a federated setting.

*Table 1.* Performance comparison on the Dream3 benchmark.

| Method | Ecoli.1 | Ecoli.2 | Yeast.1 | Yeast.2 | Yeast.3 | Average |
|---|---|---|---|---|---|---|
| | AUROC | AUROC | AUROC | AUROC | AUROC | AUROC |
| Local-dyno | 0.557 | 0.551 | 0.544 | 0.538 | 0.532 | 0.544 |
| All-dyno | 0.598 | 0.592 | 0.586 | 0.581 | 0.574 | 0.586 |
| PFDBNL | 0.531 | 0.526 | 0.521 | 0.517 | 0.512 | 0.521 |
| FDBNL | 0.586 | 0.563 | 0.558 | 0.544 | 0.528 | 0.556 |
| **Fed-CAW** | **0.606** | **0.582** | **0.573** | **0.551** | **0.546** | **0.572** |

## 5.3. Results on synthetic and real-world datasets

**Results on synthetic datasets.** Across all three synthetic suites, Fed-CAW consistently attains the lowest SHD and the highest F1-score among the federated approaches. In Synthetic I (Figure 2), where we increase the fraction and magnitude of intervention-induced perturbations, Fed-CAW maintains high F1-score and low SHD, whereas the performance of FDBNL and PFDBNL deteriorates as more short-term edges are injected into the data; this supports our design of window smoothing and backbone extraction for filtering out transient effects. In Synthetic II (Figure 3), where the total sample size is fixed but the number of clients grows, Local-dyno quickly breaks down as each client receives only a short time series, and the DBN-based federated baselines show noticeably higher SHD than Fed-CAW. By contrast, Fed-CAW remains stable across all client counts, indicating that credibility-aware aggregation effectively downweights fragile edges supported by small or noisy clients. In Synthetic III (Figure 4), where we vary the heterogeneity of sampling intervals, competing baselines suffer from substantial degradation as the maximum sampling interval increases, while Fed-CAW exhibits much milder performance loss. This confirms that aligning client-specific dynamics to a common continuous-time scale before aggregation can substantially mitigate the bias induced by heterogeneous sampling.

**Results on Dream3 networks.** For the Dream3 gene-regulatory network inference task (Table 1), Fed-CAW improves over Local-dyno and the two federated DBN baselines in AUROC averaged across networks, and slightly surpasses FDBNL despite operating under privacy constraints. All-dyno, which trains a single centralized model on pooled data, still offers an optimistic upper bound, yet the gap to Fed-CAW is modest, suggesting that credibility-aware aggregation can recover competitive regulatory structure without sharing raw gene-expression trajectories.

**Results on fMRI datasets.** For the fMRI connectivity task (Table 2), Fed-CAW approaches the centralized All-dyno reference in both F1-score and SHD, while consistently outperforming the federated baselines. This indicates that

*Table 2.* Performance comparison on fMRI benchmark.

| Method | F1-score | SHD | AUPRC | AUROC |
|---|---|---|---|---|
| Local-dyno | 0.656 | 5.60 | 0.642 | 0.720 |
| All-dyno | 0.832 | 4.20 | 0.703 | 0.825 |
| FDBNL | 0.753 | 5.80 | 0.605 | 0.735 |
| PFDBNL | 0.708 | 6.30 | 0.583 | 0.732 |
| **Fed-CAW** | **0.802** | **4.30** | **0.685** | **0.787** |

the proposed framework can recover reliable connectivity structures even when the data are distributed across subjects and cannot be directly pooled. Compared with Local-dyno and the DBN-based federated baselines, Fed-CAW achieves consistently better overall recovery performance. These results demonstrate that credibility-aware aggregation remains effective on realistic neuroimaging time series, where subject level heterogeneity and privacy constraints make direct centralized learning undesirable.

### 5.4. Ablation Studies

We conduct an ablation study to examine the contribution of the main components in Fed-CAW. As shown in Table 3, TS denotes temporal smoothing, AW denotes the joint use of time-scale alignment and credibility weighting, and DP denotes differential privacy. The full model achieves the lowest SHD and the highest F1-score among all variants, indicating that these components jointly improve structural recovery. Removing AW causes a clear degradation, showing that aggregation without time-scale alignment and credibility weighting is unreliable under heterogeneous sampling and uneven edge reliability. Further removing TS weakens the results, confirming that temporal smoothing helps suppress transient perturbations and recover stable local backbones. The variant without DP perturbation is included to isolate the effect of privacy noise. Its slight improvement over the corresponding DP variant remains clearly below the full model, suggesting that the main gains come from TS and AW, while DP introduces only a moderate utility cost.

*Table 3.* Ablation analysis on the synthetic benchmark.

| Module | | | SHD | F1-score | AUPRC | AUROC |
|---|---|---|---|---|---|---|
| TS | AW | DP | | | | |
| ✓ | ✓ | ✓ | **3.75(2.06)** | **0.83(0.10)** | **0.84(0.10)** | **0.89(0.03)** |
| ✓ | × | ✓ | 8.00(4.08) | 0.70(0.14) | 0.73(0.06) | 0.82(0.07) |
| × | × | ✓ | 8.50(3.70) | 0.67(0.14) | 0.74(0.13) | 0.80(0.05) |
| × | × | × | 7.30(3.51) | 0.69(0.13) | 0.76(0.12) | 0.83(0.09) |

## 6. Conclusion

We propose Fed-CAW for federated causal discovery from multi-client time series with temporal drift and heteroge-

neous sampling. By aggregating privatized edge evidence through credibility-aware fusion under differential privacy, Fed-CAW robustly recovers both global and personalized causal graphs. In future work, we will extend this framework to regimes with latent confounding and missing variables, and incorporate uncertainty into edge evidence.

## Acknowledgments

This work was supported by the National Natural Science Foundation of China (U24A20323 and 62376145), the Taiyuan City "Double Hundred Research Action" of the first batch project about "Leading the Charge with Open Competition" (2024TYJB0127), and Key Technology Research Project of Taihang Laboratory (THYF-KFKT-25020200).

## Impact Statement

We provide a practical way to learn dynamic causal structures from time series collected across multiple clients while keeping raw sequences local, which makes it suitable for scenarios with strict privacy and regulatory requirements. By handling temporal drift within clients and differences in sampling rates across clients, it yields more reliable causal insights for personalized decision making. Potential application domains include healthcare across hospitals and wearable devices, industrial and urban monitoring with distributed sensors, and online platforms that must evaluate time varying interventions without exposing individual level data.

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

# A. Related Work

## A.1. Causal Discovery for Time Series Data

Recently, score-based learning of directed acyclic graphs (DAGs) has been reformulated as a continuous constrained optimization problem within the Notears framework (Zheng et al., 2018). Dynotears (Pamfil et al., 2020) extends this idea to multivariate time series through a linear VAR-type model that jointly learns instantaneous and lagged graphs. NTSnotears (Sun et al., 2023) employs one-dimensional convolutional networks within a dynamic Bayesian network formulation to capture non-parametric temporal dependencies. IDYNO (Gao et al., 2022) targets non-linear dynamic Bayesian networks with both observational and interventional time series data, and TECDI (Li et al., 2023) exploits interventional temporal data to improve identifiability of causal graphs in the temporal domain. NTiCD (Absar et al., 2023) learns a shared adjacency mask for non-linear mechanisms that remain unchanged over time, Ts-causalNN (Faruque et al., 2025) uses temporally regularized neural networks to model non-linear non-stationary influences, and DyCAST (Cheng et al., 2025) adopts neural ordinary differential equations on a constrained manifold to capture causal structures that vary continuously over time. Despite these advances, the above methods are designed for centralized data collected from a single environment under a fixed sampling scheme. Even when non-stationarity or temporal drift is permitted, the causal structure is still inferred from a single data source, and thus fails to capture heterogeneity across clients or respect the privacy constraints that arise in federated settings.

## A.2. Federated CSL from Decentralized Data

Federated CSL seeks to learn a global causal graph from data stored across multiple clients without centralizing raw observations. Existing approaches can be broadly divided into score-based and continuous-optimization-based methods. Score-based methods adapt classical structure search to the federated setting. PERI (Mian et al., 2023) runs Greedy Equivalence Search (GES) locally on each client and aggregates edge proposals via a worst-case regret objective, while DARLS (Ye et al., 2024) explores the DAG space with a distributed simulated annealing scheme under communication constraints. Continuous-optimization-based federated CSL extends NOTEARS-style formulations to multi-client scenarios: Notears-ADMM (Ng & Zhang, 2022) uses ADMM to decouple optimization across clients while enforcing consensus on a shared DAG, and FedDAG (Gao et al., 2023) jointly learns a global adjacency matrix and neural causal mechanisms via federated gradient updates. These federated CSL methods do not account for the temporal dimension and therefore cannot be directly applied to time series data with temporal dependence. Recently, FDBNL and PFDBNL (Chen et al., 2026) have extended federated CSL to time-series data by learning dynamic Bayesian networks from decentralized longitudinal observations, with PFDBNL further allowing personalized client-specific structures. These methods typically assume identical sampling frequencies across clients and approximately stationary underlying dynamics, so their effectiveness is limited in practical federated time-series settings where short-lived interventions or differences in sampling mechanisms are present.

# B. Additional Theoretical Details

**Generative view and roadmap.** The section develops the theoretical underpinnings of Fed-CAW. We begin by formalizing the assumptions required for the analysis. We then motivate the unified time-scale mapping based on continuous-time reparameterization, and describe an auxiliary curve-based smoothing routine used solely for diagnostics and visualization, along with a toy example illustrating how heterogeneous sampling intervals are projected onto a common temporal grid. Next, we establish asymptotic normality for window-wise and client-level edge evidence, and characterize the null distribution of the credibility-weighted meta statistic used in global aggregation. Finally, we analyze privacy by bounding the sensitivity of the shared edge evidence and showing that the resulting Gaussian evidence mechanism satisfies an $(\varepsilon, \delta)$ differential privacy guarantee.

## B.1. Theoretical Assumptions

**Assumption B.1** (Causal sufficiency). There is no unobserved confounder acting on any pair of observed variables.

**Assumption B.2** (Cross-client null independence). For any candidate edge $e$ that is absent in the true causal structure for all clients (global null), the corresponding client-level statistics $\{z_e^{(m)}\}_{m=1}^M$ are asymptotically independent across clients as the effective sample sizes grow.

**Assumption B.3** (Bounded edge-level dependence for privacy calibration). There exist constants $\varepsilon_{\mathrm{corr}} \in (0,1)$ and

$n_{\max} < \infty$ such that, for every client $\mathcal{C}_m$, window $k$, and edge $e$,

$$\left| \hat{r}_{k,e}^{(m)} \right| \le 1 - \varepsilon_{\text{corr}}, \qquad n_{m,k} \le n_{\max},$$

where $\hat{r}_{k,e}^{(m)}$ is the (partial) correlation used in the Gaussian CI tests and $n_{m,k}$ is the effective sample size in window $k$.

## B.2. Continuous-Time Reparameterization for a Unified Time-Scale Mapping

**Lemma B.4** (Unique generator from a stable transition matrix). *Let $F \in \mathbb{R}^{d \times d}$ be a matrix whose eigenvalues lie strictly inside the unit disk. Then there exists a unique real matrix $G \in \mathbb{R}^{d \times d}$ with eigenvalues having negative real parts such that $F = \exp(G)$, and $G = \log(F)$ where $\log(\cdot)$ denotes the principal matrix logarithm.*

*Proof.* Let $F \in \mathbb{R}^{d \times d}$ have all eigenvalues strictly inside the open unit disk.

First, we establish the existence of the principal logarithm. Since the spectrum of $F$ does not intersect the non-positive real axis $(-\infty, 0]$, the principal matrix logarithm $\log(F)$ is well-defined and analytic in a neighborhood of $F$; see, e.g., Higham (2008, Thm. 11.2). More concretely, if $\Gamma$ is a simple closed contour enclosing the spectrum of $F$ and avoiding $(-\infty, 0]$, then

$$\log(F) = \frac{1}{2\pi i} \int_\Gamma \log(z) \, (zI - F)^{-1} \, dz,$$

where $\log(z)$ is the principal branch of the complex logarithm with branch cut on $(-\infty, 0]$. By the functional calculus for matrices, this satisfies $\exp(\log(F)) = F$.

Next, we characterize the spectrum of $\log(F)$. Because $F$ is real and $\Gamma$ can be chosen symmetric with respect to the real axis, the above integral yields a real matrix $\log(F)$ (Higham (2008, Sec. 11.7)). Let $\lambda_1, \ldots, \lambda_d$ be the eigenvalues of $F$, with $\lambda_j = r_j e^{i\theta_j}$ where $r_j > 0$ and $\theta_j \in (-\pi, \pi)$. The eigenvalues of $\log(F)$ are $\log(\lambda_j)$, where

$$\log(\lambda_j) = \log r_j + i\theta_j.$$

Since $|\lambda_j| = r_j < 1$, we have $\log r_j < 0$, so every eigenvalue of $G := \log(F)$ has strictly negative real part.

Finally, we argue uniqueness under a spectral strip constraint. Suppose $G_1, G_2$ are (complex) matrices such that $\exp(G_1) = \exp(G_2) = F$ and all eigenvalues of $G_1, G_2$ lie in the strip

$$\mathcal{S} = \{\lambda \in \mathbb{C} : -\pi < \Im(\lambda) < \pi\}.$$

Then, by Higham (2008, Thm. 11.9), $G_1 = G_2 = \log(F)$, i.e., the principal logarithm is the unique logarithm whose spectrum is contained in $\mathcal{S}$. In our case, the eigenvalues of $G = \log(F)$ all have negative real part, hence they lie in $\mathcal{S}$. If $G'$ is any other (real) matrix with $\exp(G') = F$ and eigenvalues having negative real parts, then its spectrum also lies in $\mathcal{S}$, and therefore $G' = G = \log(F)$. $\qquad\square$

**Proof of Proposition 4.1 (Approximate alignment).** We show that the estimated generator coincides with the true generator under the exact continuous-time model, and that the aligned transition matrices are stable under mild heterogeneity.

*Proof.* Fix a window index $k$. By assumption, for each client $\mathcal{C}_m$, the companion matrix $\Phi_k^{(m)}$ has all eigenvalues strictly inside the unit disk.

For statement (i), Lemma B.4, applied to $F = \Phi_k^{(m)}$, implies the existence of a unique real matrix $H^{(m)} = \log(\Phi_k^{(m)})$ whose eigenvalues have negative real parts and satisfy $\exp(H^{(m)}) = \Phi_k^{(m)}$. On the other hand, the assumption that the discrete-time dynamics in window $k$ for client $\mathcal{C}_m$ are generated by some continuous-time matrix $G_k^{(m)}$ implies

$$\Phi_k^{(m)} = \exp(G_k^{(m)} \Delta_m).$$

Thus $G_k^{(m)} \Delta_m$ is a (real) logarithm of $\Phi_k^{(m)}$. By uniqueness of the principal logarithm in Lemma B.4, we must have

$$G_k^{(m)} \Delta_m = \log(\Phi_k^{(m)}).$$

Therefore

$$\hat{G}_k^{(m)} = \frac{1}{\Delta_m} \log(\Phi_k^{(m)}) = G_k^{(m)},$$

and substituting into the definition of the aligned transition matrix gives

$$\widetilde{\Phi}_k^{(m)} = \exp(\hat{G}_k^{(m)} \Delta_{\mathrm{ref}}) = \exp(G_k^{(m)} \Delta_{\mathrm{ref}}),$$

which proves item (i).

For statement (ii), let $\mathcal{K}$ be a compact set of matrices that contains all $G_k^{(m)}$ and $G_k^\star$. Consider the map

$$T : \mathcal{K} \to \mathbb{R}^{d \times d}, \qquad T(A) = \exp(A \Delta_{\mathrm{ref}}).$$

The matrix exponential is analytic in $A$, hence $T$ is Lipschitz on $\mathcal{K}$, i.e., there exists a constant $L_k > 0$ such that

$$\left\| \exp(A \Delta_{\mathrm{ref}}) - \exp(B \Delta_{\mathrm{ref}}) \right\| \le L_k \|A - B\| \quad \text{for all } A, B \in \mathcal{K}. \tag{25}$$

For completeness, one can obtain an explicit $L_k$ using the integral representation

$$\exp(A \Delta_{\mathrm{ref}}) - \exp(B \Delta_{\mathrm{ref}}) = \int_0^1 \frac{d}{ds} \exp\big((1-s)B\Delta_{\mathrm{ref}} + sA\Delta_{\mathrm{ref}}\big) \, ds$$

$$= \int_0^1 \exp\big((1-s)B\Delta_{\mathrm{ref}} + sA\Delta_{\mathrm{ref}}\big)(A - B)\Delta_{\mathrm{ref}} \, ds.$$

Taking norms and using submultiplicativity gives

$$\left\| \exp(A \Delta_{\mathrm{ref}}) - \exp(B \Delta_{\mathrm{ref}}) \right\| \le \Delta_{\mathrm{ref}} \sup_{C \in \mathcal{K}} \| \exp(C\Delta_{\mathrm{ref}}) \| \cdot \|A - B\|,$$

so (25) holds with $L_k = \Delta_{\mathrm{ref}} \sup_{C \in \mathcal{K}} \| \exp(C\Delta_{\mathrm{ref}}) \|$.

Applying (25) with $A = G_k^{(m)}$ and $B = G_k^\star$, and using part (i), we obtain

$$\left\| \widetilde{\Phi}_k^{(m)} - \exp(G_k^\star \Delta_{\mathrm{ref}}) \right\| = \left\| \exp(G_k^{(m)} \Delta_{\mathrm{ref}}) - \exp(G_k^\star \Delta_{\mathrm{ref}}) \right\| \le L_k \|G_k^{(m)} - G_k^\star\| \le L_k \, \delta_k.$$

This proves item (ii). $\qquad\qquad\qquad\qquad\qquad\qquad\qquad\qquad\qquad\qquad\qquad\qquad\qquad\qquad\qquad\qquad\qquad\square$

### B.3. Auxiliary Curve-Based Time-Scale Mapping (Optional Smoothing)

We present an auxiliary curve-based procedure used solely for optional smoothing and diagnostic visualization. This step is applied only for inspection purposes and does not affect the edge-level statistics or any quantities used in the main Fed-CAW aggregation.

**Description.** After generator-based alignment onto the reference step $\Delta_{\mathrm{ref}}$ in Section 3.3, client $\mathcal{C}_m$ is equipped with lag matrices $\{\bar{\mathbf{A}}_\ell^{(m)}\}_{\ell=1}^{L_{\mathrm{ref}}}$ on the common grid. For a directed edge $e = (i \to j)$, we collect the aligned lag coefficients into

$$\bar{a}_{\ell,e}^{(m)} := \big| \bar{\mathbf{A}}_\ell^{(m)}(j,i) \big|, \qquad \tau_\ell := \ell \, \Delta_{\mathrm{ref}}, \quad \ell = 1, \ldots, L_{\mathrm{ref}}. \tag{26}$$

To obtain a smooth description of how the strength of $e$ decays over physical time, we approximate the discrete profile $\{\bar{a}_{\ell,e}^{(m)}\}_{\ell=1}^{L_{\mathrm{ref}}}$ by a simple nonincreasing exponential curve

$$f_e^{(m)}(\tau; \theta_e^{(m)}) = \alpha_e^{(m)} \exp\big(-\beta_e^{(m)} \tau\big), \qquad \alpha_e^{(m)} \ge 0, \; \beta_e^{(m)} \ge 0, \tag{27}$$

where $\theta_e^{(m)} = (\alpha_e^{(m)}, \beta_e^{(m)})$ are edge-specific parameters.

The parameters are obtained via a nonnegative least-squares fit

$$\hat{\theta}_e^{(m)} = \arg \min_{\alpha \ge 0, \, \beta \ge 0} \sum_{\ell=1}^{L_{\mathrm{ref}}} \Big( \bar{a}_{\ell,e}^{(m)} - \alpha \exp(-\beta \tau_\ell) \Big)^2, \tag{28}$$

optionally supplemented with a small $\ell_2$ penalty on $(\alpha, \beta)$ in very low-sample regimes.

To reconstruct a lag-wise profile on the reference grid, we evaluate the fitted curve at the same time points and restore the dominant sign of the edge. Let

$$s_e^{(m)} = \text{sign}\Big(\sum_{\ell=1}^{L_{\text{ref}}} \bar{\mathbf{A}}_\ell^{(m)}(j, i)\Big), \tag{29}$$

and define the auxiliary coefficients

$$\tilde{A}_{\ell,e}^{(m)} = s_e^{(m)} f_e^{(m)}(\tau_\ell; \hat{\theta}_e^{(m)}), \qquad \ell = 1, \dots, L_{\text{ref}}. \tag{30}$$

In our implementation, the generator-based coefficients and the resulting edge statistics $(z_e^{(m)}, \kappa_e^{(m)})$ remain the primary quantities used for federated aggregation. The auxiliary curve $\{\tilde{A}_{\ell,e}^{(m)}\}_\ell$ is only used to obtain a smoother description of the lag profile of $e$ (for instance, when defining time-consistency indicators across clients or for visual inspection), and does not alter the aggregated edge statistics in Fed-CAW.

**Consistency of two-point exponential extrapolation.** To clarify why this auxiliary step is compatible with the generator-based view, we consider the idealized noiseless case where the physical-time effect of edge $e$ on client $\mathcal{C}_m$ follows an exact exponential decay

$$w_e^{(m)}(\tau) = c_e^{(m)} \exp\big(-r_e^{(m)}\tau\big), \qquad c_e^{(m)} > 0, \ r_e^{(m)} \geq 0. \tag{31}$$

Assume that, after generator-based alignment, client $\mathcal{C}_m$ observes two lag magnitudes

$$\bar{a}_{\ell_1,e}^{(m)} = w_e^{(m)}(\tau_{\ell_1}), \qquad \bar{a}_{\ell_2,e}^{(m)} = w_e^{(m)}(\tau_{\ell_2}),$$

with $0 < \tau_{\ell_1} < \tau_{\ell_2}$.

**Proposition B.5.** *In the above setting, restricting the fit* (27)–(28) *to the two points* $(\tau_{\ell_1}, \bar{a}_{\ell_1,e}^{(m)})$ *and* $(\tau_{\ell_2}, \bar{a}_{\ell_2,e}^{(m)})$ *exactly recovers the decay parameters,*

$$\hat{r}_e^{(m)} = \frac{1}{\tau_{\ell_2} - \tau_{\ell_1}} \log \frac{\bar{a}_{\ell_1,e}^{(m)}}{\bar{a}_{\ell_2,e}^{(m)}}, \qquad \hat{c}_e^{(m)} = \bar{a}_{\ell_1,e}^{(m)} \exp\big(\hat{r}_e^{(m)}\tau_{\ell_1}\big), \tag{32}$$

*and the fitted curve matches the underlying coefficient at any lag* $\tau$:

$$f_e^{(m)}(\tau; \hat{\theta}_e^{(m)}) = w_e^{(m)}(\tau) = c_e^{(m)} \exp\big(-r_e^{(m)}\tau\big). \tag{33}$$

*In particular, for a target lag* $\tau^\star = 2\Delta_{\text{ref}}$, *the extrapolated value* $f_e^{(m)}(\tau^\star; \hat{\theta}_e^{(m)})$ *coincides with the true coefficient at physical delay* $\tau^\star$.

*Proof.* From the exponential form,

$$\bar{a}_{\ell_1,e}^{(m)} = c_e^{(m)} e^{-r_e^{(m)}\tau_{\ell_1}}, \qquad \bar{a}_{\ell_2,e}^{(m)} = c_e^{(m)} e^{-r_e^{(m)}\tau_{\ell_2}}.$$

Taking the ratio gives

$$\frac{\bar{a}_{\ell_1,e}^{(m)}}{\bar{a}_{\ell_2,e}^{(m)}} = e^{r_e^{(m)}(\tau_{\ell_2} - \tau_{\ell_1})},$$

so

$$r_e^{(m)} = \frac{1}{\tau_{\ell_2} - \tau_{\ell_1}} \log \frac{\bar{a}_{\ell_1,e}^{(m)}}{\bar{a}_{\ell_2,e}^{(m)}}.$$

Substituting back into $\bar{a}_{\ell_1,e}^{(m)} = c_e^{(m)} e^{-r_e^{(m)}\tau_{\ell_1}}$ yields

$$c_e^{(m)} = \bar{a}_{\ell_1,e}^{(m)} \exp(r_e^{(m)}\tau_{\ell_1}).$$

Thus $(\hat{c}_e^{(m)}, \hat{r}_e^{(m)}) = (c_e^{(m)}, r_e^{(m)})$, and $f_e^{(m)}(\tau; \hat{\theta}_e^{(m)}) = w_e^{(m)}(\tau)$ for all $\tau$, including $\tau^\star = 2\Delta_{\text{ref}}$. $\qquad \square$

**B.4. Toy Example: Mapping Three Clients to a Unified Time Scale**

We illustrate how clients with different sampling intervals can be mapped to a shared time scale while recovering consistent edge coefficients.

**Step 1: True dynamics on the reference time scale.** Consider two variables $X_t^1$ and $X_t^2$ evolving at the reference step $\Delta_{\mathrm{ref}} = 1$ according to a linear VAR(1) model,

$$\begin{bmatrix} X_t^1 \\ X_t^2 \end{bmatrix} = A \begin{bmatrix} X_{t-1}^1 \\ X_{t-1}^2 \end{bmatrix} + \varepsilon_t, \qquad A = \begin{bmatrix} 0.8 & 0.2 \\ 0.1 & 0.7 \end{bmatrix}. \tag{34}$$

Thus, at the reference time scale, $X_{t-1}^1$ has a lag-one autoregressive effect $X_{t-1}^1 \to X_t^1$ with coefficient 0.8, and $X_{t-1}^2$ has a lag-one cross effect $X_{t-1}^2 \to X_t^1$ with coefficient 0.2.

**Step 2: What each client observes at its own sampling interval.** Suppose all clients share the same underlying continuous-time generator $G = \log(A)$, but sample at different intervals. For the reference process, the $k$-step transition matrix is $A^k = \exp(kG)$. We consider three clients:

**Client $\mathcal{C}_1$ ($\Delta_1 = 1$).** Client $\mathcal{C}_1$ records every time step. Its discrete model coincides with Eq. (34) with transition matrix

$$A^{(1)} = A = \begin{bmatrix} 0.80 & 0.20 \\ 0.10 & 0.70 \end{bmatrix}.$$

Here the lag-one effect of $X_{t-1}^2$ on $X_t^1$ is 0.20.

**Client $\mathcal{C}_2$ ($\Delta_2 = 2$).** Client $\mathcal{C}_2$ only records every second time point $t = 0, 2, 4, \ldots$. Eliminating the unobserved intermediate states yields a VAR(1) model at step size 2 with transition matrix

$$A^{(2)} = A^2 = \begin{bmatrix} 0.66 & 0.30 \\ 0.15 & 0.51 \end{bmatrix}.$$

In this representation, the two-step autoregressive effect from $X_{t-2}^1$ to $X_t^1$ is 0.66, and the two-step cross effect from $X_{t-2}^2$ to $X_t^1$ becomes 0.30.

**Client $\mathcal{C}_3$ ($\Delta_3 = 3$).** Client $\mathcal{C}_3$ records every third time point $t = 0, 3, 6, \ldots$ and therefore sees

$$A^{(3)} = A^3 = \begin{bmatrix} 0.62 & 0.34 \\ 0.17 & 0.45 \end{bmatrix},$$

so the three-step self effect from $X_{t-3}^1$ to $X_t^1$ is 0.62 and the three-step cross effect from $X_{t-3}^2$ to $X_t^1$ is 0.34.

All three clients observe different discrete-time coefficients, even though they share the same underlying causal mechanism.

**Step 3: Mapping all clients back to the common time scale.** To align these models to the reference step $\Delta_{\mathrm{ref}} = 1$, we interpret each $A^{(m)}$ as arising from the same continuous-time generator $G$:

$$A^{(m)} = \exp(\Delta_m G), \qquad m \in \{1, 2, 3\}.$$

We then recover $G$ on each client by

$$\widehat{G}^{(m)} = \frac{1}{\Delta_m} \log\big(A^{(m)}\big),$$

where $\log(\cdot)$ denotes the principal matrix logarithm.

For the numerical example above, the generator computed from $\mathcal{C}_1$ is

$$\widehat{G}^{(1)} = \log(A) \approx \begin{bmatrix} -0.241 & 0.270 \\ 0.135 & -0.376 \end{bmatrix}.$$

Client $\mathcal{C}_2$ obtains

$$\log\big(A^{(2)}\big) = \log(A^2) \approx \begin{bmatrix} -0.482 & 0.540 \\ 0.270 & -0.753 \end{bmatrix},$$

so dividing by its sampling interval $\Delta_2 = 2$ gives

$$\widehat{G}^{(2)} = \frac{1}{2}\log(A^{(2)}) \approx \begin{bmatrix} -0.241 & 0.270 \\ 0.135 & -0.376 \end{bmatrix} = \widehat{G}^{(1)}.$$

Similarly, client $\mathcal{C}_3$ has

$$\log(A^{(3)}) = \log(A^3) \approx \begin{bmatrix} -0.724 & 0.810 \\ 0.405 & -1.129 \end{bmatrix}, \qquad \widehat{G}^{(3)} = \frac{1}{3}\log(A^{(3)}) \approx \begin{bmatrix} -0.241 & 0.270 \\ 0.135 & -0.376 \end{bmatrix},$$

again matching $\widehat{G}^{(1)}$ up to numerical rounding.

Mapping back to the reference step $\Delta_{\text{ref}} = 1$ uses

$$\widetilde{A}^{(m)} = \exp(\Delta_{\text{ref}}\widehat{G}^{(m)}) = \exp(\widehat{G}^{(m)}),$$

which yields

$$\widetilde{A}^{(1)} = \widetilde{A}^{(2)} = \widetilde{A}^{(3)} \approx \begin{bmatrix} 0.80 & 0.20 \\ 0.10 & 0.70 \end{bmatrix} = A.$$

In particular, the lag-one self effect $X_{t-1}^1 \to X_t^1$ is returned to the common coefficient $0.80$ at the reference scale, even though it appears as $0.80$ (client 1), $0.66$ (client 2), and $0.62$ (client 3) when expressed at step sizes $\Delta_1, \Delta_2, \Delta_3$.

Likewise, the lag-one cross effect $X_{t-1}^2 \to X_t^1$ is mapped to the unified coefficient $0.20$ in $A$, instead of the heterogeneous values $0.20, 0.30$, and $0.34$ observed in $A^{(1)}, A^{(2)}, A^{(3)}$ before alignment. Consequently, all clients share the same edge set $\{X^1 \to X^1, \ X^2 \to X^1, \ X^1 \to X^2, \ X^2 \to X^2\}$, and the generator-based mapping makes the corresponding coefficients directly comparable on the common time scale.

Intuitively, Proposition 4.1 analyzes the alignment step under a mildly heterogeneous scenario where each client has its own generator $G_k^{(m)}$, but all generators in the same window lie in a bounded neighborhood of some reference dynamics $G_k^\star$. We do not require the generators, or the resulting graphs, to coincide exactly across clients.

Overall, Fed-CAW is designed to extract, for each client, a temporally stable backbone graph that filters out short-term local interventions, and to fuse these backbones across clients in a credibility-aware fashion so as to recover a partially shared global structure together with personalized deviations.

### B.5. Asymptotic Properties of Local Edge Evidence

We establish asymptotic normality of the window-wise Fisher $z$-statistic for each candidate edge, which serves as the building block for client-level aggregation.

**Lemma B.6** (Asymptotic normality of window-wise edge evidence). *Fix a client $\mathcal{C}_m$, a window $k$, and an edge $e = (i, j, \ell)$. Assume that, within $\mathcal{W}_k^{(m)}$, the process follows the linear time-series SEM in Eq. (2) with an acyclic instantaneous matrix, and that $\{Z_{k,t}^{(m)}\}_{t \in \mathcal{W}_k^{(m)}}$ is mean-zero, sub-Gaussian, and independent over time. Let $n_{m,k} \to \infty$ as $T_m \to \infty$.*

*Let $\hat{r}_{k,e}^{(m)}$ be the sample partial correlation for testing $e$ and define the Fisher $z$-statistic*

$$z_{k,e}^{(m)} = \sqrt{n_{m,k} - 3} \, \frac{1}{2} \log \frac{1 + \hat{r}_{k,e}^{(m)}}{1 - \hat{r}_{k,e}^{(m)}}.$$

*Then $z_{k,e}^{(m)} \xrightarrow{d} \mathcal{N}(\mu_e^{(m)}, 1)$, where $\mu_e^{(m)} = 0$ under the local null and $\mu_e^{(m)} \neq 0$ otherwise.*

*Proof.* Fix a client $\mathcal{C}_m$, a window $k$, and a candidate edge $e = (i, j, \ell)$. Within this window, consider the augmented vector

$$Z_t = \big(X_t^{(m)\top}, X_{t-1}^{(m)\top}, \ldots, X_{t-L}^{(m)\top}\big)^\top \in \mathbb{R}^{d(L+1)},$$

for $t \in \mathcal{W}_k^{(m)}$.

First, under the assumed linear SEM Eq. (2), acyclicity of the instantaneous coefficient matrix $W_k^{(m)}$, and mean-zero, sub-Gaussian, time-independent exogenous noise $\{Z_{k,t}^{(m)}\}_{t \in \mathcal{W}_k^{(m)}}$, the process $\{Z_t\}_{t \in \mathcal{W}_k^{(m)}}$ is strictly stationary with finite second moments and satisfies standard weak dependence conditions (as in vector autoregressive processes). Let $\Sigma = \mathbb{E}[Z_t Z_t^\top]$ denote its covariance matrix and $\hat{\Sigma}$ the sample covariance based on the $n_{m,k}$ effective observations in window $k$. By a multivariate central limit theorem for weakly dependent sub-Gaussian sequences, we have

$$\sqrt{n_{m,k}} \, \mathrm{vec}(\hat{\Sigma} - \Sigma) \xrightarrow{d} \mathcal{N}(0, \Omega),$$

for some finite covariance matrix $\Omega$.

Next, we express the relevant partial correlation as a smooth function of $\Sigma$. Let $\mathcal{P}_i$ denote the set of indices corresponding to the parents of $X_i^t$ in the true DAG (including both instantaneous and lagged parents). Let $u$ be the coordinate index of $X_j^{t-\ell}$ and $v$ that of $X_i^t$ in $Z_t$. The population partial correlation between $X_j^{t-\ell}$ and $X_i^t$ conditional on $X_{\mathcal{P}_i \setminus \{j\}}$ can be expressed in terms of the inverse covariance $\Theta = \Sigma^{-1}$ as

$$r_e^{(m)} = -\frac{\Theta_{uv}}{\sqrt{\Theta_{uu}\Theta_{vv}}}.$$

The map $\Sigma \mapsto \Theta = \Sigma^{-1}$ is smooth on the set of positive definite matrices, and the map $\Theta \mapsto r_e^{(m)}$ is rational with non-zero denominator near the true $\Theta$. Therefore the composition

$$g : \Sigma \mapsto r_e^{(m)} = g(\Sigma)$$

is smooth in a neighborhood of the true covariance matrix. Let $\hat{r}_{k,e}^{(m)} = g(\hat{\Sigma})$ be the sample partial correlation.

Then, by the multivariate delta method applied to $g$, we obtain

$$\sqrt{n_{m,k}}\big(\hat{r}_{k,e}^{(m)} - r_e^{(m)}\big) \xrightarrow{d} \mathcal{N}(0, \tau_e^2),$$

where

$$\tau_e^2 = \nabla g(\Sigma)^\top \Omega \, \nabla g(\Sigma)$$

is finite and strictly positive.

Next, we apply a second delta method via the Fisher transform. Define

$$\tilde{z}_{k,e}^{(m)} = h\big(\hat{r}_{k,e}^{(m)}\big), \qquad h(r) = \frac{1}{2}\log\frac{1+r}{1-r}.$$

The function $h(r)$ is analytic on $(-1, 1)$ and hence smooth in a neighborhood of the true partial correlation $r_e^{(m)}$. A one-dimensional delta method yields

$$\sqrt{n_{m,k}}\big(\tilde{z}_{k,e}^{(m)} - z_e^{(m)}\big) \xrightarrow{d} \mathcal{N}\big(0, v_e^2\big),$$

where $z_e^{(m)} = h(r_e^{(m)})$ and

$$v_e^2 = \big(h'(r_e^{(m)})\big)^2 \tau_e^2, \qquad h'(r) = \frac{1}{1-r^2}.$$

Finally, for Gaussian data, the exact asymptotic variance of the Fisher transformed (partial) correlation is

$$\mathrm{Var}\big(\tilde{z}_{k,e}^{(m)}\big) \approx \frac{1}{n_{m,k}-3},$$

independently of $r_e^{(m)}$; see, e.g., (Anderson, 2003). Thus the classical Fisher $z$-statistic is

$$z_{k,e}^{(m)} = \sqrt{n_{m,k}-3}\,\tilde{z}_{k,e}^{(m)} = \sqrt{n_{m,k}-3}\,\frac{1}{2}\log\frac{1+\hat{r}_{k,e}^{(m)}}{1-\hat{r}_{k,e}^{(m)}},$$

which satisfies

$$z_{k,e}^{(m)} \xrightarrow{d} \mathcal{N}(\mu_e^{(m)}, 1),$$

with $\mu_e^{(m)} = 0$ if $r_e^{(m)} = 0$ (no edge) and $\mu_e^{(m)} \neq 0$ otherwise. This coincides with the statement of the lemma. $\square$

## B.6. Asymptotic Normality of Aggregated Client-Level Evidence

Deterministic window aggregation preserves asymptotic normality and yields a client-level evidence variable suitable for server-side fusion.

**Lemma B.7** (Asymptotic normality of client-level aggregated evidence). *Fix a client $\mathcal{C}_m$ and an edge $e$. Let $\{z_{k,e}^{(m)}\}_{k=1}^{K_m}$ be the window-wise Fisher z statistics from Lemma B.6, and define the client-level aggregated statistic as in Eq. (4):*

$$z_e^{(m)} = \sum_{k=1}^{K_m} \omega_{k,e}^{(m)} z_{k,e}^{(m)}, \qquad \sum_{k=1}^{K_m} \omega_{k,e}^{(m)} = 1,$$

*where the weights $\omega_{k,e}^{(m)}$ are deterministic. Then, as $\min_k n_{m,k} \to \infty$,*

$$z_e^{(m)} \xrightarrow{d} \mathcal{N}\big(\mu_e^{(m)}, \sigma_e^{2,(m)}\big),$$

*where*

$$\mu_e^{(m)} = \sum_{k=1}^{K_m} \omega_{k,e}^{(m)} \mu_{k,e}^{(m)}, \qquad \sigma_e^{2,(m)} = \sum_{k=1}^{K_m} \big(\omega_{k,e}^{(m)}\big)^2,$$

*and $\mu_{k,e}^{(m)}$ are the window-wise limiting means from Lemma B.6. In particular, under the local null for client $\mathcal{C}_m$, $\mu_e^{(m)} = 0$.*

*Proof.* By Lemma B.6, each $z_{k,e}^{(m)}$ is asymptotically normal with mean $\mu_{k,e}^{(m)}$ and variance 1. Since the weights $\{\omega_{k,e}^{(m)}\}_k$ are deterministic and $\sum_k \omega_{k,e}^{(m)} = 1$, $z_e^{(m)}$ is a finite linear combination of asymptotically Gaussian variables. Standard arguments (e.g., Cramér–Wold) imply that $z_e^{(m)}$ is asymptotically Gaussian with mean $\mu_e^{(m)} = \sum_k \omega_{k,e}^{(m)} \mu_{k,e}^{(m)}$ and variance $\sigma_e^{2,(m)} = \sum_k (\omega_{k,e}^{(m)})^2$. Under the local null, all $\mu_{k,e}^{(m)} = 0$, hence $\mu_e^{(m)} = 0$. $\qquad \square$

## B.7. Proof of Theorem 4.2 (Meta statistic)

Theorem 4.2 derives the asymptotic null distribution of the credibility-weighted meta statistic used for global aggregation.

*Proof.* For each client $\mathcal{C}_m$, the edge-level statistic

$$z_e^{(m)} = \sum_k \omega_{k,e}^{(m)} z_{k,e}^{(m)}$$

is a finite linear combination of asymptotically normal variables. Under the local null, Lemma B.6 implies $z_{k,e}^{(m)} \Rightarrow \mathcal{N}(0,1)$, and hence (by Lemma B.7) $z_e^{(m)} \Rightarrow \mathcal{N}(0,1)$.

Define

$$S_e = \sum_{m=1}^{M} w_e^{(m)} z_e^{(m)}.$$

Under the global null hypothesis that edge $e$ is absent for all clients, we have $\mu_e^{(m)} = 0$ for all $m$. Moreover, by Assumption B.2, the statistics $\{z_e^{(m)}\}_{m=1}^{M}$ are asymptotically independent and each converges in distribution to $\mathcal{N}(0,1)$. Hence, for large samples,

$$S_e \approx \sum_{m=1}^{M} w_e^{(m)} Z_m, \qquad Z_m \sim \mathcal{N}(0,1) \text{ i.i.d.}$$

The mean and variance of $S_e$ are then

$$\mathbb{E}[S_e] = 0, \qquad \text{Var}(S_e) = \sum_{m=1}^{M} (w_e^{(m)})^2.$$

A finite linear combination of independent Gaussian random variables is Gaussian, so

$$S_e \xrightarrow{d} \mathcal{N}\left(0, \sum_{m=1}^{M}(w_e^{(m)})^2\right),$$

and therefore the normalized meta-statistic

$$Z_e^{\text{meta}} = \frac{S_e}{\sqrt{\sum_{m=1}^{M}(w_e^{(m)})^2}}$$

converges in distribution to $\mathcal{N}(0,1)$ under the global null.

Under fixed non-zero local effects, each $z_e^{(m)}$ has mean $\mu_e^{(m)} \neq 0$ and variance approximately one. The mean and variance of $S_e$ are

$$\mathbb{E}[S_e] = \sum_{m=1}^{M} w_e^{(m)} \mu_e^{(m)}, \qquad \text{Var}(S_e) = \sum_{m=1}^{M}(w_e^{(m)})^2.$$

Thus

$$\mathbb{E}[Z_e^{\text{meta}}] = \frac{\sum_{m=1}^{M} w_e^{(m)} \mu_e^{(m)}}{\sqrt{\sum_{m=1}^{M}(w_e^{(m)})^2}}.$$

If there exists at least one client $\mathcal{C}_m$ with $\mu_e^{(m)} > 0$ and $w_e^{(m)} > 0$, this mean is strictly positive. The more clients with large positive $\mu_e^{(m)}$ and large credibility weights $w_e^{(m)}$, the larger the aggregated mean, corresponding to a higher signal-to-noise ratio at the meta level.

The variance of $Z_e^{\text{meta}}$ remains asymptotically one because the weights are deterministic and the $z_e^{(m)}$ retain unit variance and asymptotic independence. Hence, under the alternative, $Z_e^{\text{meta}}$ remains asymptotically normal with unit variance but with a strictly positive mean that increases with the aggregated signal strength. $\square$

### B.8. Privacy Analysis: Sensitivity Bounds and the Gaussian Mechanism for Edge Evidence

We establish privacy for sharing client-side edge evidence in two steps. First, we show that the stacked evidence vector has a finite $\ell_2$-sensitivity under Assumption B.3. This yields an explicit sensitivity bound $\Delta_2$. We then apply the Gaussian mechanism to this query and invoke a standard calibration to obtain $(\varepsilon, \delta)$-differential privacy.

**Proof of Proposition 4.3 (Bounded sensitivity).** The key observation is that Assumption B.3 enforces a uniform bound on each window-wise Fisher statistic. Since the client-level statistic is a convex combination of window-wise quantities, it inherits the same bound, which directly implies a finite sensitivity for the stacked vector.

*Proof.* Under Assumption B.3, for every client $\mathcal{C}_m$, window $k$ and edge $e$ we have

$$\left|\hat{r}_{k,e}^{(m)}\right| \leq 1 - \varepsilon_{\text{corr}}, \qquad n_{m,k} \leq n_{\max}.$$

Recall that the window-wise Fisher statistic is

$$z_{k,e}^{(m)} = \sqrt{n_{m,k} - 3}\, \frac{1}{2} \log \frac{1 + \hat{r}_{k,e}^{(m)}}{1 - \hat{r}_{k,e}^{(m)}}.$$

Consider the scalar function

$$g(r,n) = \sqrt{n-3}\, \frac{1}{2} \log \frac{1+r}{1-r}, \qquad r \in (-1,1),\ n \geq 3.$$

On the compact set

$$\mathcal{K} = \left\{(r,n) : |r| \leq 1 - \varepsilon_{\text{corr}},\ 3 \leq n \leq n_{\max}\right\},$$

the function $g$ is continuous, hence bounded. Define

$$B_z := \max_{(r,n)\in\mathcal{K}} |g(r,n)| < \infty.$$

Then for all $m, k, e$, $|z_{k,e}^{(m)}| \le B_z$.

The aggregated statistic for client $\mathcal{C}_m$ and edge $e$ is

$$z_e^{(m)} = \sum_k \omega_{k,e}^{(m)} z_{k,e}^{(m)}, \qquad \sum_k \omega_{k,e}^{(m)} = 1, \; \omega_{k,e}^{(m)} \ge 0.$$

Since $z_{k,e}^{(m)} \in [-B_z, B_z]$ for all $k$ and the weights form a convex combination, we also have

$$|z_e^{(m)}| \le B_z \quad \text{for all } m, e.$$

Now consider two neighboring local datasets $D^{(m)}, D^{(m)\prime}$ for client $\mathcal{C}_m$. Let $z_e^{(m)}$ and $z_e^{(m)\prime}$ denote the corresponding aggregated statistics. For each edge $e$ we have

$$|z_e^{(m)} - z_e^{(m)\prime}| \le 2B_z.$$

Let $\mathcal{E}_m$ denote the set of candidate edges on client $\mathcal{C}_m$, and stack all edge-level statistics into

$$z^{(m)}(D^{(m)}) = \big\{ z_e^{(m)} \big\}_{e\in\mathcal{E}_m} \in \mathbb{R}^{|\mathcal{E}_m|}.$$

We obtain

$$\big\| z^{(m)}(D^{(m)}) - z^{(m)}(D^{(m)\prime}) \big\|_2^2 = \sum_{e\in\mathcal{E}_m} \big( z_e^{(m)} - z_e^{(m)\prime} \big)^2 \le |\mathcal{E}_m|\, (2B_z)^2.$$

Thus

$$\big\| z^{(m)}(D^{(m)}) - z^{(m)}(D^{(m)\prime}) \big\|_2 \le 2B_z \sqrt{|\mathcal{E}_m|} =: \Delta_2 < \infty,$$

which proves the claimed bounded-sensitivity property. $\qquad\square$

**Proof of Theorem 4.5 (Gaussian evidence mechanism).**

*Proof.* Let $z^{(m)}(D^{(m)}) \in \mathbb{R}^d$ denote the stacked vector of all edge-level evidence statistics for client $\mathcal{C}_m$, where $d = |\mathcal{E}_m|$. By Proposition 4.3, $z^{(m)}(\cdot)$ has $\ell_2$-sensitivity at most $\Delta_2$, i.e., for any pair of neighboring datasets $D^{(m)}, D^{(m)\prime}$ we have

$$\big\| z^{(m)}(D^{(m)}) - z^{(m)}(D^{(m)\prime}) \big\|_2 \le \Delta_2.$$

The mechanism can be written as

$$\mathcal{M}(D^{(m)}) = z^{(m)}(D^{(m)}) + \xi, \qquad \xi \sim \mathcal{N}(0, \sigma^2 I_d).$$

This is exactly the Gaussian mechanism applied to the query $z^{(m)}$ with sensitivity $\Delta_2$. It is a standard result (see, e.g., Dwork & Roth (2014, Thm. 3.22)) that if

$$\sigma \ge \frac{\Delta_2}{\varepsilon} \sqrt{2 \log \frac{1.25}{\delta}},$$

then $\mathcal{M}$ is $(\varepsilon, \delta)$-differentially private for any pair of neighboring datasets with $\ell_2$-sensitivity at most $\Delta_2$. Since this condition holds by Proposition 4.3, the stated choice of $\sigma$ guarantees client-level $(\varepsilon, \delta)$-DP. $\qquad\square$

## C. Detailed Algorithmic Description of Fed-CAW

This section summarizes the complete Fed-CAW procedure in pseudocode. Algorithm 1 details the client-side local structure learning and the construction of edge-level credibility scores under client-level differential privacy. Algorithm 2 then shows how the server aggregates these privatized statistics through credibility weighted meta analysis and computes edge-level sharing probabilities for personalization. Finally, Algorithm 3 presents the outer coordination loop that alternates between the client and server steps and uses a stopping rule driven by stability considerations.

---

**Algorithm 1** Client-side local Structure Learning and Credibility Scores Construction

1: **Input:** client set $\{\mathcal{C}_m\}_{m=1}^M$ with local time series $D^{(m)} = \{\mathbf{X}_t^{(m)}\}_{t=1}^{T_m}$, sampling intervals $\{\Delta_m\}_{m=1}^M$; lag order $L$, window length $G$ and stride $s$, local significance level $\alpha_{\text{loc}}$, reference step size $\Delta_{\text{ref}}$; DP parameters $(\varepsilon, \delta)$
2: **Output:** local graphs $\{\mathcal{G}_{\text{local}}^{(m)}\}_{m=1}^M$ and privatized signatures $\{\tilde{s}_e^{(m)}\}_{m=1}^M$
3: **for** each client $\mathcal{C}_m$ **do**
4:     **Step 1: Window smoothing and within-client local aggregation**
5:     construct sliding windows $\{\mathcal{W}_k^{(m)}\}_{k=1}^{K_m}$ on $D^{(m)}$ with length $G$ and stride $s$
6:     jointly optimize the cross-window smoothed SEM objective on $\{\mathcal{W}_k^{(m)}\}$ to obtain coefficients $\{(\mathbf{W}_k^{(m)}, \{\mathbf{A}_{k,\ell}^{(m)}\}_{\ell=1}^L)\}_{k=1}^{K_m}$
7:     **for** each window $k = 1, \ldots, K_m$ **do**
8:         compute window-wise Fisher-$z$ statistics $\{z_{k,e}^{(m)}\}_{e \in \mathcal{E}}$ based on $(\mathbf{W}_k^{(m)}, \{\mathbf{A}_{k,\ell}^{(m)}\}_{\ell=1}^L)$
9:     **end for**
10:     aggregate window-wise statistics into client-level statistics $z_e^{(m)}$ as in Eq. (4)
11:     perform significance testing at level $\alpha_{\text{loc}}$ and obtain the local graph $\mathcal{G}_{\text{local}}^{(m)} \leftarrow (V, \mathcal{E}_{\text{loc}}^{(m)})$
12:     **Step 2: Credibility score computation on selected edges**
13:     **for** each edge $e \in \mathcal{E}_{\text{loc}}^{(m)}$ **do**
14:         compute temporal stability credibility $\kappa_e^{(m)}$ using the support set in Eq. (5) and frequency definition in Eq. (6)
15:         compute the generator $\hat{G}_k^{(m)}$ and aligned transition $\tilde{\Phi}^{(m)}$ for each window in Eq. (11) - (3.3) using $\Delta_m$ and $\Delta_{\text{ref}}$
16:         extract the aligned lag profile $\tilde{\mathbf{a}}_{e,k}^{(m)}$ in Eq. (12) and compute temporal consistency credibility $\rho_e^{(m)}$ in Eq. (15)
17:     **Step 3: DP perturbation of $z_e$ and signature construction**
18:     sample $\xi_{z,e}^{(m)} \sim \mathcal{N}(0, \sigma_z^2)$ with $\sigma_z$ calibrated from $(\varepsilon, \delta)$ in the DP analysis
19:     set $\tilde{z}_e^{(m)} \leftarrow z_e^{(m)} + \xi_{z,e}^{(m)}$
20:     form the signature $\tilde{s}_e^{(m)} \leftarrow (\tilde{z}_e^{(m)}, \kappa_e^{(m)}, \rho_e^{(m)})$
21:     **end for**
22:     send $\{\tilde{s}_e^{(m)}\}$ and $\Delta_m$ to the server
23: **end for**

---

**Algorithm 2** Server-side Credibility-Weighted Aggregation and Sharing-Probability Personalization

1: **Input:** privatized edge signatures $\{\tilde{s}_e^{(m)}\}_{m=1}^M$, global significance level $\alpha_{\text{glob}}$
2: **Output:** global graph $\mathcal{G}^{\text{glob}} = (V, \mathcal{E}_{\text{glob}})$, sharing probability $\mathbf{Q}_e$ and personalized graphs $\{\mathcal{G}_{\text{local}}^{(m)}\}_{m=1}^M$
3: **Step 1: Credibility-weighted global aggregation**
4: **for** each edge $e$ **do**
5:     parse $\tilde{s}_e^{(m)} = (\tilde{z}_e^{(m)}, \kappa_e^{(m)}, \rho_e^{(m)})$ for all $m$ with reports on edge $e$
6:     compute credibility weights and normalized coefficients $\{\alpha_e^{(m)}\}_{m=1}^M$ using $\kappa_e^{(m)}$ and $\rho_e^{(m)}$
7:     compute the meta statistic $z_{\text{meta},e} \leftarrow \sum_{m=1}^M \alpha_e^{(m)} \tilde{z}_e^{(m)}$ in Eq. (17)
8:     perform a global significance test at level $\alpha_{\text{glob}}$ and obtain the global graph $\mathcal{G}_{\text{global}} \leftarrow (V, \mathcal{E}_{\text{glob}})$
9: **end for**
10: **Step 2: Sharing probabilities and personalized optimization**
11: **for** each edge $e \in \mathcal{E}_{\text{glob}}$ **do**
12:     compute sharing probability $Q_e$ using breadth and directional agreement in Eq. (18)
13: **end for**
14: broadcast $(\mathcal{G}^{\text{glob}}, \mathbf{Q}_e)$ to all clients
15: **for** each client $\mathcal{C}_m$ **do**
16:     update local instantaneous and lagged parameters using edge-level sharing probabilities, e.g.,
17:     $\mathbf{W}^{(m)} \leftarrow \mathbf{Q}_W \odot \mathbf{W}^{\text{glob}} + (1 - \mathbf{Q}_W) \odot \mathbf{W}_{\text{local}}^{(m)}$
18:     $\mathbf{A}_\ell^{(m)} \leftarrow \mathbf{Q}_{A,\ell} \odot \mathbf{A}_\ell^{\text{glob}} + (1 - \mathbf{Q}_{A,\ell}) \odot \mathbf{A}_{\ell,\text{local}}^{(m)}, \quad \ell = 1, \ldots, L$
19:     obtain personalized graph $\mathcal{G}_{\text{local}}^{(m)}$ from the updated parameters if desired
20: **end for**

---

**Algorithm 3** Overall Fed-CAW coordination loop with stability-based stopping

---

1: **Input:** maximum number of rounds $R_{\max}$, stability patience $R_{\text{stab}} = 3$, initial global graph $\mathcal{G}_{(0)}^{\text{glob}} = (V, \emptyset)$
2: **Output:** final global graph $\mathcal{G}_{(r^\star)}^{\text{glob}}$ and personalized graphs $\{\mathcal{G}_{\text{local}}^{(m)}\}_{m=1}^{M}$
3: set $r \leftarrow 1$, stab_cnt $\leftarrow 0$
4: **while** $r \leq R_{\max}$ **and** stab_cnt $< R_{\text{stab}}$ **do**
5:    **Client phase:** each client runs Algorithm 1 to produce privatized signatures $\{\tilde{s}_e^{(m)}\}$
6:    **Server phase:** server runs Algorithm 2 to obtain $\mathcal{G}_{(r)}^{\text{glob}}$ and $\{\mathcal{G}_{\text{local}}^{(m)}\}_{m=1}^{M}$
7:    compare $\mathcal{G}_{(r)}^{\text{glob}}$ with $\mathcal{G}_{(r-1)}^{\text{glob}}$
8:    **if** $\mathcal{G}_{(r)}^{\text{glob}} = \mathcal{G}_{(r-1)}^{\text{glob}}$ **then**
9:      stab_cnt $\leftarrow$ stab_cnt $+ 1$
10:   **else**
11:     stab_cnt $\leftarrow 0$
12:   **end if**
13:   server broadcasts $(\mathcal{G}_{(r)}^{\text{glob}}, \{Q_e\})$ and updated parameters to all clients for the next round
14:   $r \leftarrow r + 1$
15: **end while**
16: let $r^\star \leftarrow r - 1$ and output $\mathcal{G}_{(r^\star)}^{\text{glob}}$ and $\{\mathcal{G}_{\text{local}}^{(m)}\}_{m=1}^{M}$

---

The three algorithms provide complementary views of the same federated procedure. Algorithm 1 describes the quantities computed locally on each client and the privatized signatures transmitted to the server. Algorithm 2 explains how the server aggregates these transmitted statistics into global anchors and edge-level sharing probabilities. Algorithm 3 then specifies how the global anchors are fed back to clients for subsequent personalized updates across communication rounds. This makes that raw time series remain on local clients throughout the procedure, while only compact edge-level summaries are communicated to the server.

## D. Experimental Details and Additional Results

The section provides supplementary details on the experimental settings, datasets, and baseline methods used in Section 5.

### D.1. Supplementary Experimental Setup and Parameters

**Synthetic settings and default configurations.** Table 4 summarizes the key problem attributes in our synthetic experiments, together with the set of values explored in each sensitivity study and the default value used unless stated otherwise.

*Table 4.* Synthetic experiment configurations: ablated values and defaults.

| Attribute | Ablated values | Default |
|---|---|---|
| # variables $d$ | $\{2, 4, 8, 16, 32\}$ | 8 |
| # clients $M$ | $\{2, 4, 8, 16, 32\}$ | 16 |
| series length per client $T_m$ | $\{100, 200, 500\}$ | 200 |
| SVAR order $L$ | $\{1, 2\}$ | 1 |
| fraction of time under intervention | $\{0.0, 0.1, 0.2, 0.4\}$ | 0.4 |
| perturbation magnitude $\Delta$ | $\{0.0, 0.2, 0.4, 0.6\}$ | 0.4 |
| sampling heterogeneity $\Delta_{\max}/\Delta_{\min}$ | $\{1, 2, 4, 8\}$ | 4 |

*Synthetic experiment parameters.* Local learning on each client: $\alpha_{\text{loc}} = 0.01$, $\lambda_W = \lambda_A = 0.1$, $K_{\text{alt}} = 6$, $K_{\text{gd}} = 120$, learning rates $\eta_W = \eta_A = 0.01, 0.01$, and temporal smoothness regularization coefficients $\gamma_W = \gamma_A = 0.1$, window length $G = T_m/4$, step size $s = T_m/8$. Server aggregation: $\alpha_{\text{glob}} = 0.05$. Privacy: client level Gaussian mechanism with $(\varepsilon, \delta) = (50, 10^{-5})$.

**Real-world datasets.** Table 5 reports basic statistics of the real-world datasets used in Section 5.2, including the number of variables, the number of time points, and how we emulate a multi-client federated setting.

*Benchmark experiment parameters.* Local learning on each client: $\alpha_{\text{loc}} = 0.2$, $\lambda_W = \lambda_A = 0.2$, $K_{\text{alt}} = 6$, $K_{\text{gd}} = 120$, learning rates $\eta_W = \eta_A = 0.01, 0.01$, and temporal smoothness coefficients $r_W = r_A = 0.1$, window length $G = T_m/2$, step size $s = T_m/4$. Server aggregation: $\alpha_{\text{glob}} = 0.2$. Privacy: client level Gaussian mechanism with $(\varepsilon, \delta) = (50, 10^{-5})$.

*Table 5.* Real-world datasets and federated emulation protocols.

| Dataset | # variables $d$ | # time points per series | # clients $M$ | Notes |
|---|---|---|---|---|
| Dream3 Size100 | 100 | 189 | 5 | multiple experimental conditions split across clients |
| fMRI (simulated) | $\{5, 10, 15\}$ | $\{200, 1200\}$ | 50 | subjects treated as clients |

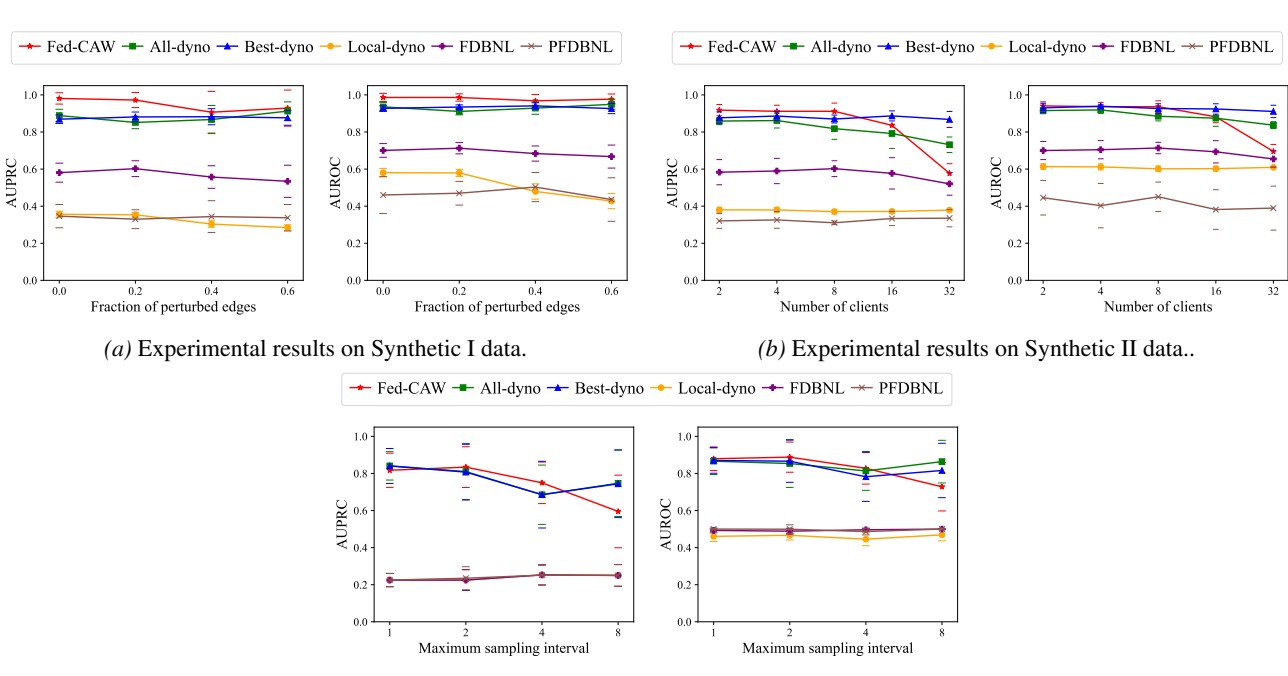

*(a)* Experimental results on Synthetic I data.

*(b)* Experimental results on Synthetic II data..

*(c)* Experimental results on Synthetic III data.

*Figure 5.* AUROC and AUPRC on synthetic datasets across experimental settings.

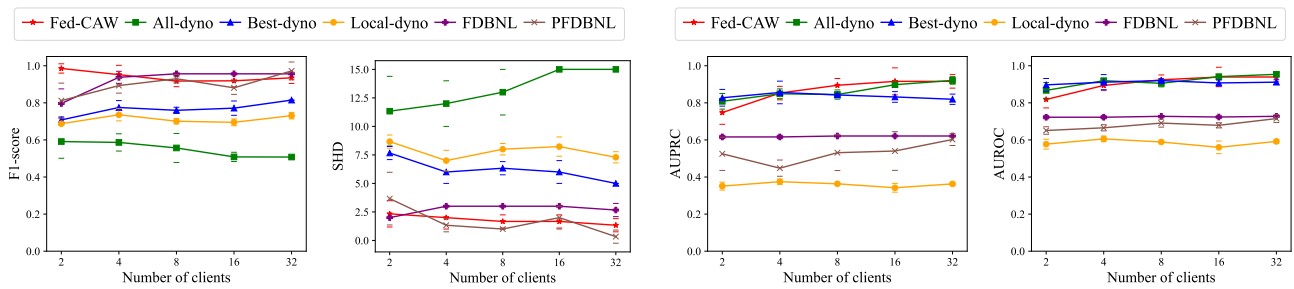

*Figure 6.* Experimental results under temporal drift with varying numbers of clients.

## D.2. Additional Evaluation Metrics and Experimental Findings

The main text reports SHD and F1-score for the three synthetic suites in Section 5.3. We additionally report AUROC and AUPRC under the same settings, which summarize the ranking quality of real-valued edge scores.

**Synthetic I: AUROC and AUPRC under temporal drift.** Figure 5a reports AUROC and AUPRC as the fraction of perturbed edges increases. Across all drift levels, Fed-CAW attains the highest scores and closely tracks or exceeds the centralized Dynotears baselines, while Local-dyno and the two federated DBN variants deteriorate more noticeably as interventions become stronger. This is consistent with the SHD/F1-score trends, and shows that temporal smoothing and credibility-aware weighting help preserve ranking quality even under pronounced drift.

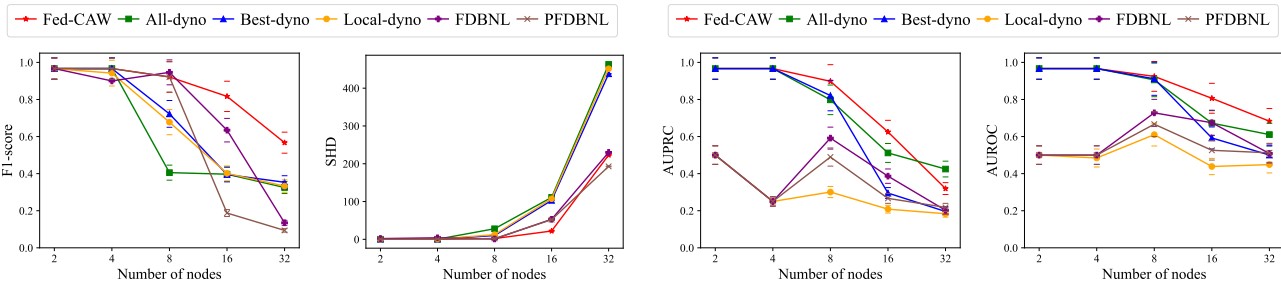

*Figure 7.* Experimental results under temporal drift with varying numbers of variables.

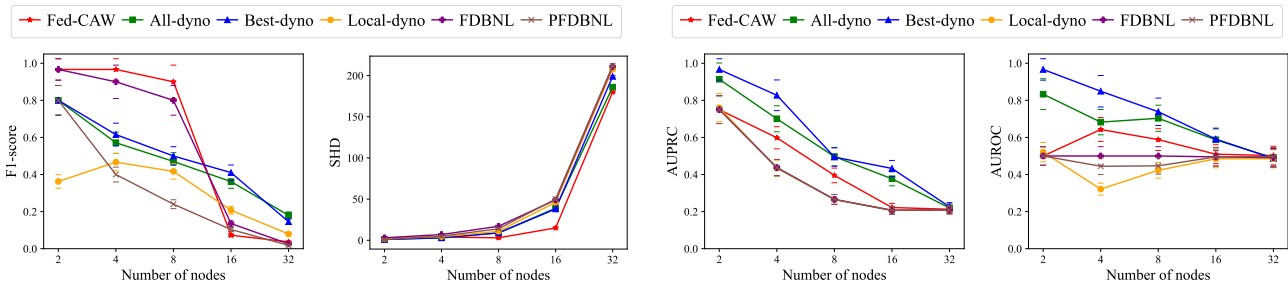

*Figure 8.* Experimental results under heterogeneous sampling with varying numbers of variables.

**Synthetic II: AUROC and AUPRC under client scaling.** Figure 5b reports AUROC and AUPRC with a fixed total number of observations while increasing the number of clients. As per-client sequences shorten, AUROC and AUPRC of Local-dyno drop substantially, and FDBNL/PFDBNL remain clearly below Fed-CAW. Fed-CAW maintains strong performance across all client counts and stays competitive with centralized Dynotears variants, and the AUROC/AUPRC trends closely match the SHD and F1-score results, with Fed-CAW providing the most reliable structures among federated methods as the federation scales.

**Synthetic III: AUROC and AUPRC under heterogeneous sampling.** Figure 5c shows AUROC and AUPRC as the maximum sampling interval grows, i.e., sampling heterogeneity increases. Methods that rely on a coarsened common grid suffer marked degradation in both metrics, especially PFDBNL, while Fed-CAW exhibits only mild losses and consistently achieves the best ranking performance. Together with the SHD/F1-score results, this highlights the benefit of continuous-time alignment for stabilizing cross-client evidence under heterogeneous sampling.

In Section 5.3, we report three primary synthetic settings where we change one factor at a time and keep the remaining settings fixed. Here we include additional experiments under fixed drift or fixed sampling heterogeneity regimes, and vary either the number of clients or the problem dimensionality to further examine robustness of structural recovery.

**Temporal drift with varying numbers of clients.** Figure 6 reports F1-score, SHD, AUROC, and AUPRC when the temporal drift setting from Synthetic I is kept fixed while the number of clients $M$ increases under an approximately fixed total number of observations. As per–client sequences become shorter, Local-dyno degrades markedly in both F1-score and AUPRC and its SHD increases, and FDBNL/PFDBNL remain clearly below Fed-CAW on all four metrics. Fed-CAW maintains high F1-score and AUPRC and low SHD across all values of $M$, and stays competitive with centralized Dynotears variants in AUROC, suggesting that credibility–aware aggregation together with backbone smoothing effectively counters the loss of sample size on individual clients.

**Temporal drift with varying numbers of variables.** Figure 7 shows the same temporal–drift setting while varying the number of variables $d$. All methods deteriorate as $d$ grows, but the decline is much stronger for the federated DBN baselines, whose F1-score drops quickly and whose SHD and ranking metrics worsen. Fed-CAW exhibits a slower loss of accuracy, keeping a visible gap over FDBNL and PFDBNL in both SHD/F1-score and AUROC/AUPRC, and remaining close to centralized Dynotears methods for moderate dimensions.

**Heterogeneous sampling with varying numbers of variables.** Figure 8 considers heterogeneous sampling with fixed number of clients and fixed heterogeneity ratio $\Delta_{\max}/\Delta_{\min}$ while increasing $d$. F1-score and AUPRC decrease and SHD rises for all methods as the dimension grows, with the DBN baselines showing the fastest degradation. Fed-CAW retains better F1-score and AUPRC and lower SHD than the federated DBN methods, and remains competitive with centralized Dynotears for smaller and medium $d$, which is consistent with the benefit of continuous–time alignment combined with credibility–aware aggregation under heterogeneous sampling.

