# OpenReview forum: "Credibility-Aware Weighting Federated Causal Discovery for Time Series"
_ICML.cc/2026/Conference — ICML 2026 regular_

### Official Review · Reviewer_2feG · 2026-03-10

**Soundness:** 3
**Presentation:** 3
**Significance:** 3
**Originality:** 3
**Overall Recommendation:** 5
**Confidence:** 3

**Summary:**

This paper tackles a tough problem in causal discovery: how do you map out cause-and-effect in federated systems when every client is running on a different clock? We're talking about heterogeneous sampling frequencies and causal mechanisms that actually drift over time. To solve this, the authors introduce Fed-CAW, a framework built around 'credibility-aware' aggregation.
Locally, it uses sliding-window SVAR models with some smart regularization to keep things smooth. The real heavy lifting happens next: they use matrix logarithms to align these different time scales into a single, unified view. They then score each causal edge based on how stable and consistent it is across time. Finally, the server aggregates everything using a weighted meta-analysis that respects differential privacy.

**Compliance With Llm Reviewing Policy:**

Affirmed.

**Final Justification:**

I would say temporal drift is the novelty in federated causal discovery.
The authors use known methods of reliability weighting, and continuous-time alignment. The novelty lies mainly in their combination.
The results look sound

**Key Questions For Authors:**

mentioned above in the weaknesses

**Limitations:**

The paper discusses future extensions, but does not sufficiently address several limitations such as sensitivity to hyperparameters, robustness to model misspecification, and the realism of theoretical assumptions like causal sufficiency and cross-client independence.

**Strengths And Weaknesses:**

Strengths:
1. Most federated causal discovery papers assume everyone is on the same clock, which just isn't true in practice. This paper's focus on heterogeneous sampling and drifting mechanisms makes it much more applicable to in-the-wild data.
2. The way the authors have layered the components—from sliding-window SEMs to the matrix-log alignment—is very intuitive. It doesn't feel like a collection of random tricks; it's a well-thought-out workflow.
3. I really liked the edge credibility idea. By looking at how stable a causal link is over time and checking its lag consistency, the model has a built-in way to filter out the transient edges that usually plague these kinds of studies.

Weaknesses:
1. The method combines multiple modules: window smoothing, time-scale alignment, credibility weighting, and differential privacy
However, ablation studies are limited, making it difficult to understand which components contribute most to performance.
2. The credibility metrics (κ and ρ) are mathematically defined but their practical interpretation and sensitivity to hyperparameters (e.g., thresholds, window sizes) are not fully explored.
3. Some theoretical results rely on assumptions such as causal sufficiency and independence across clients under the null hypothesis, which may not hold in realistic federated environments.
4. Limited number of baselines; If not mistaken, e.g. FedDAG could be very relevant.

---

> ### Author Rebuttal · Authors · 2026-03-31
>
> We thank the reviewer for the insightful questions and helpful remarks. We hope that our responses below address the issues raised.
>
> **W1:** The method combines multiple modules: window smoothing, time-scale alignment, credibility weighting, and differential privacy. However, ablation studies are limited, making it difficult to understand which components contribute most to performance.
>
> **R1:** To quantify the contribution of each module, we conducted an ablation study, and the results are shown below. Here, TS denotes temporal smoothing, AW denotes alignment and weighted aggregation, and DP denotes privacy noise. Since alignment and weighted aggregation are coupled in our framework and both build on temporally smoothed local estimates, we treat them as a single module in the ablation. The full model achieves the best performance on synthetic benchmark, and removing each component leads to degraded performance. Overall, these results clarify the contribution of each module within the sequential ablation design: removing AW leads to the most immediate degradation, removing TS causes additional performance loss, and DP maintains a favorable balance between privacy and utility. We will clarify this point in the revised manuscript.
>
> |Dataset|TS|AW|DP|SHD|F1|AUPRC|AUROC|
> |---|---|---|---|---:|---:|---:|---:|
> |Synthetic|✓|✓|✓|**3.75(2.06)**|**0.83(0.10)**|**0.84(0.10)**|**0.89(0.03)**|
> |Synthetic|✓|×|✓|8.00(4.08)|0.70(0.14)|0.73(0.06)|0.82(0.07)|
> |Synthetic|×|×|✓|8.50(3.70)|0.67(0.14)|0.74(0.13)|0.80(0.05)|
> |Synthetic|×|×|×|7.30(3.51)|0.69(0.13)|0.76(0.12)|0.83(0.09)|
>
> **W2:** The credibility metrics $\kappa$ and $\rho$ are mathematically defined but their practical interpretation and sensitivity to hyperparameters (e.g., thresholds, window sizes) are not fully explored.
>
> **R2:** In our framework, $\kappa$ reflects the persistence and stability of an edge across windows, whereas $\rho$ reflects the consistency of its aligned lag profile. Accordingly, $\kappa$ captures within-client temporal reliability, while $\rho$ captures cross-window consistency after alignment. An edge receives a high credibility weight only when both quantities are high. The thresholding effect is controlled by the same significance level $\alpha$ used in the local procedure. We further evaluate the sensitivity to the window parameters $G$ and $s$, as shown below, where Mean $\kappa$ and Mean $\rho$ denote the average credibility scores over the selected edge set. The results suggest that a moderate window length with a relatively small stride gives the best overall performance and higher credibility scores. We will clarify this point in the revised manuscript.
>
> |$G$|$s$|windows$K_m$|Mean $\kappa$|Mean $\rho$|SHD|F1|
> |---|---|---:|---:|---:|---:|---:|
> |$T_m/8$|$G/4$|28|0.78(0.11)|0.84(0.07)|6.33(2.08)|0.76(0.10)|
> |$T_m/8$|$G/2$|14|0.81(0.07)|0.87(0.05)|5.30(2.65)|0.78(0.04)|
> |$T_m/4$|$G/4$|12|0.91(0.08)|0.94(0.03)|**4.33(1.45)**|**0.87(0.06)**|
> |$T_m/4$|$G/2$|7|0.89(0.03)|0.92(0.02)|4.75(1.60)|0.83(0.07)|
>
> **W3:** Some theoretical results rely on assumptions such as causal sufficiency and independence across clients under the null hypothesis, which may not hold in realistic federated environments.
>
> **R3:** These assumptions are standard working assumptions introduced to support the theoretical analysis, rather than claims that every real federated deployment satisfies them exactly. In particular, causal sufficiency is a common idealization in causal discovery that has been adopted in many prior studies, and it can still be a reasonable approximation in well-instrumented federated settings such as industrial process monitoring. We also acknowledge that confounding may arise in practice, and extending our framework to such cases is an important direction for future work. Moreover, in the horizontal federated setting considered here, cross-client independence under the null provides a natural approximation for deriving the null distribution of the meta-statistic without requiring identical client dynamics.
>
> **W4:** Limited number of baselines; If not mistaken, e.g. FedDAG could be very relevant.
>
> **R4:** FedDAG is relevant to federated causal discovery, but it is originally designed for static independent samples rather than multivariate time series with temporal dependence and lagged dynamics. We therefore did not include it as a direct baseline in the main comparison. To provide a reference point, we constructed a simplified comparison on a synthetic benchmark by converting each client time series into lag-augmented static samples under a controlled setting. The results in the table below show that our method remains competitive, suggesting that FedDAG is less suitable for federated settings with temporal drift and heterogeneous sampling frequencies.
>
> |Method|SHD|F1|AUPRC|AUROC|
> |---|---:|---:|---:|---:|
> |FedDAG|7.33(1.52)|0.63(0.15)|0.78(0.10)|0.73(0.05)|
> |Ours|4.60(1.49)|0.84(0.05)|0.87(0.04)|0.90(0.05)|

---

> > ### Author Rebuttal · Reviewer_2feG · 2026-04-04
> >
> > The authors provided new ablation study, which resolved my questions. I increase my score

---

> > > ### Author Response · Authors · 2026-04-05
> > >
> > > Thank you for your acknowledging of our work and for raising score. We also sincerely appreciate your time and effort in reviewing our paper.

---

### Official Review · Reviewer_GVrT · 2026-03-12

**Soundness:** 3
**Presentation:** 3
**Significance:** 3
**Originality:** 3
**Overall Recommendation:** 5
**Confidence:** 4

**Summary:**

This paper introduces Fed-CAW, a credibility aware weighting federated causal discovery method for multivariate time series. The method combines local structure learning per client with credibility aggregation using differential privacy in order to learn a global causal graph without sharing any raw data. The paper tests their method's effectiveness on synthetic and real datasets and compared to SOTA baselines.

**Compliance With Llm Reviewing Policy:**

Affirmed.

**Final Justification:**

Strong paper with high significance, originality, soundness and clear presentation. My minor concerns around surrounding privacy budget and parameter sensitivity were addressed by the authors, so I raised my score.

**Key Questions For Authors:**

1.	Are there any assumptions that must be imposed for the input multivariate time series? For example, is there a maximum time series length $T_m$ or any constraints/assumptions surrounding the frequency (e.g., how often traces are sampled) of the traces?
2.	How sensitive to parameters is the method (eg like stride s and window length G)?
3.	Were any utility vs privacy budget experiments conducted, and if so does the model still perform well at more realistic privacy budgets, e.g., epsilon <= 10?

**Limitations:**

No. Would be helpful to include a limitations section to discuss any drawbacks (e.g., perhaps with the utility vs privacy budget trade offs) or related to handling traces with missing data or latent or unobservable confounders.

**Strengths And Weaknesses:**

Significance and Originality: This paper looks at the important area of causal discovery for multivariate time series in private federated settings which has applications to many real-world use cases. It presents a nontrivial and meaningful advancement in the state of the art by developing a causal discovery method that can handle temporal drift and different client sampling frequencies and incorporates differential privacy. The method performs comparably or better than previous baselines.

Presentation: The paper is well written, structured logically, easy to follow, with clear graphics and figures. Mathematical definitions are precise and understandable.

Soundness: Overall this paper appears technically sound, with valid formulation and theoretical analyses. One thing that is missing is a description of any assumptions on the input client time series - see question 1. Moreover, as is common with causal discovery and federated learning mechanisms, there are a lot of parameters that must be set and that the model may be sensitive to. It is unclear how much effect these have on the model – see question 2.

The one area of concern I have is surrounding the differential privacy component. As shown in Appendix C.1, the privacy budget is set quite high (50, 10^-5) for all of the experiments which is unlikely to provide strong enough privacy guarantees in real world use cases. In addition, notably missing is an in depth privacy analysis such as evaluation about the effect of different privacy budgets vs utility performance trade-offs. If provided, this would strengthen the experimental results. See question 3.

---

> ### Author Rebuttal · Authors · 2026-03-31
>
> We thank the reviewer for the thoughtful feedback and valuable suggestions. We hope that the following clarifications resolve the main concerns and better explain our methodology and experiments.
>
> **Q1:**  Are there any assumptions that must be imposed for the input multivariate time series? For example, is there a maximum time series length $T_m$ or any constraints/assumptions surrounding the frequency (e.g., how often traces are sampled) of the traces?
>
> **R1:** We thank the reviewer for this important question. Our framework does not impose overly restrictive assumptions on the input multivariate time series, but it requires several basic conditions to support local estimation and temporal alignment. In particular, our framework does not impose a hard upper bound on the local sequence length $T_m$. In practice, $T_m$ only needs to be large enough to support sliding-window estimation, i.e., to form at least one valid window given the lag order $L$ and window length $G$.
>
> Regarding sampling frequency, our framework assumes that the sampling frequency is consistent within each client, but it does not require all clients to follow the same sampling frequency. Instead, clients may have different sampling intervals $\Delta_m$, which is one of the main motivations of our method. We will clarify this more explicitly in the revised manuscript.
>
> **Q2:** How sensitive to parameters is the method (eg like stride $s$ and window length $G$)?
>
> **R2:** We thank the reviewer for this insightful question. In our framework, $G$ and $s$ determine the balance between temporal adaptivity and estimation stability. Smaller values improve sensitivity to short-lived mechanism drift, but may also lead to noisier local estimates. Larger values enhance estimation stability, but may smooth out transient changes. Importantly, our method mitigates over-reliance on a single window partition by aggregating edge evidence across windows and weighting it with the credibility terms $\kappa$ and $\rho$, which favor edges that remain stable over time and consistent after temporal alignment.
> | $G$ | $s$ | windows $K_m$ | SHD | F1 | AUPRC | AUROC |
> |---|---|---:|---:|---:|---:|---:|
> | $T_m/8$ | $G/4$ | 28 | 6.33(2.08) | 0.76(0.10) | 0.77(0.09) | 0.86(0.06) |
> | $T_m/8$ | $G/2$ | 14 | 5.30(2.65) | 0.78(0.04) | 0.76(0.10) | 0.83(0.08) |
> | $T_m/4$ | $G/4$ | 12 | **4.33(1.45)** | **0.87(0.06)** | **0.85(0.05)** | **0.89(0.06)** |
> | $T_m/4$ | $G/2$ | 7 | 4.75(1.60) | 0.83(0.07) | 0.81(0.09) | 0.84(0.05) |
> | $T_m/2$ | $G/4$ | 5 | 5.67(1.53) | 0.74(0.05) | 0.75(0.06) | 0.82(0.04) |
> | $T_m/2$ | $G/2$ | 3 | 7.33(3.06) | 0.63(0.12) | 0.66(0.13) | 0.76(0.09) |
>
> As shown above, the best performance is attained at $G=T_m/4$ and $s=G/4$. This suggests that an appropriately finer-grained windowing scheme is beneficial, but making the windows even finer does not necessarily yield further improvement. Overall, the results suggest that the best performance is achieved when temporal adaptivity and estimation stability are properly balanced. We will clarify this point in the revised manuscript.
>
> **Q3:** Were any utility vs privacy budget experiments conducted, and if so does the model still perform well at more realistic privacy budgets, e.g., $\varepsilon <= 10$?
>
> **R3:** We thank the reviewer for this important question. We evaluate our method under multiple privacy budgets, and the results are shown below. Overall, the results indicate a gradual between privacy and utility: as the privacy budget becomes smaller, performance degrades smoothly rather than collapsing abruptly. Moreover, the method remains effective under practically relevant privacy budgets, such as $\varepsilon \leq 10$, indicating that it preserves strong utility under moderate privacy protection. We will make this point clearer in the revised manuscript.
>
> | $\varepsilon$ | Synthetic SHD | Synthetic F1 | Synthetic AUPRC | Synthetic AUROC | fMRI SHD | fMRI F1 | fMRI AUPRC | fMRI AUROC |
> |---|---:|---:|---:|---:|---:|---:|---:|---:|
> | 1  | 5.80 (1.35) | 0.82 (0.04) | 0.83 (0.03) | 0.82 (0.06) | 5.82 | 0.752 | 0.639 | 0.738 |
> | 5  | 5.30 (0.48) | 0.87 (0.02) | 0.85 (0.04) | 0.90 (0.03) | 5.10 | 0.773 | 0.657 | 0.752 |
> | 10 | 4.70 (1.26) | 0.91 (0.05) | 0.89 (0.06) | 0.92 (0.05) | 4.70 | 0.781 | 0.674 | 0.773 |
> | 20 | 4.60 (1.10) | 0.92 (0.04) | 0.91 (0.04) | 0.95 (0.02) | 4.50 | 0.792 | 0.683 | 0.784 |

---

> > ### Author Rebuttal · Reviewer_GVrT · 2026-03-31
> >
> > Thanks for your detailed response and answers to my questions. My concerns have been alleviated and I have raised my score.

---

> > > ### Author Response · Authors · 2026-04-01
> > >
> > > Thank you for your recognition of our work and for raising the score. We also sincerely thank you for your time and effort in reviewing our paper.

---

### Official Review · Reviewer_3kKz · 2026-03-16

**Soundness:** 3
**Presentation:** 2
**Significance:** 2
**Originality:** 3
**Overall Recommendation:** 4
**Confidence:** 2

**Summary:**

The paper proposes a privacy-preserving federated method for time-series causal discovery that explicitly handles both temporal drift within clients and mismatched sampling rates across clients by weighting edges according to their credibility. The authors show the effectiveness of their approach with theoretical support for time-scale alignment and formal DP guarantees. They validate the method on synthetic and real-world datasets.

**Compliance With Llm Reviewing Policy:**

Affirmed.

**Final Justification:**

The rebuttal addressed most of my concerns, I'd maintain my positive score.

**Key Questions For Authors:**

I'm not an expert in time series and causal discovery, but for the privacy part, I have several questions for the authors:

- For the DP part, who is responsible for the noise addition? The server or the client?
- Do you aim to achieve local DP or global DP?
- Does your design rely on any DP properties? Is DP just a layer that makes your framework privacy-preserving?
- Can you analyze the complexity of your framework? Will the runtime be practical in real-world applications?
- Can you illustrate if the assumptions you made in Appendix B are realistic in practice?

**Limitations:**

I don't see negative societal impact.

**Strengths And Weaknesses:**

## Strengths:
- The setting the paper studies is a realistic setting.
- The credibility-weighted aggregation is intuitive.
- The framework is privacy-aware.

## Weaknesses:
- The setup can be better explained.
- The assumptions need justifications.

---

> ### Author Rebuttal · Authors · 2026-03-31
>
> We sincerely thank the reviewer for the careful reading and constructive feedback. We hope that the responses below clarify the raised concerns and address the questions from the reviewer.
>
> **Q1:** For the DP part, who is responsible for the noise addition? The server or the client?
>
> **R1:** In our framework, noise is added locally at each client. Specifically, each client perturbs its local edge-evidence vector with Gaussian noise and transmits only the privatized signature to the server. The server never has access to the raw local edge evidence or the raw time series data.
>
> **Q2:** Do you aim to achieve local DP or global DP?
>
> **R2:** Our framework is closer to local DP than to a centralized global DP. Concretely, each client releases a privatized edge-evidence vector by adding Gaussian noise before transmission, where the noise scale is calibrated according to the sensitivity bound and the target privacy budget $(\varepsilon,\delta)$.
>
> **Q3:** Does your design rely on any DP properties? Is DP just a layer that makes your framework privacy-preserving?
>
> **R3:** We thank the reviewer for this insightful question. Our design does not rely on special properties of differential privacy, since the core components of our method are introduced for robust federated causal discovery rather than derived from DP itself. In this sense, DP is not what makes the causal criterion work. However, DP is also not merely an additional privacy-preserving layer. Since the server only receives noisy transmitted summaries and performs aggregation directly on these privatized statistics, the aggregation rule must remain robust to the added noise. In this context, we introduce credibility-aware weighting to preserve utility by assigning greater weight to edge evidence that remains stable over time and consistent across clients.
>
> **Q4:** Can you analyze the complexity of your framework? Will the runtime be practical in real-world applications?
>
> **R4:** The computational complexity of our framework can be characterized by two stages in each communication round: client-side local updates and server-side aggregation. On the client side, since local learning and edge-level summary construction are performed over $K_m$ sliding windows, the per-round complexity is $O\left(K_m I_{\mathrm{loc}} G (L+1)d^2\right)$, where $I_{\mathrm{loc}}$ is the number of local optimization steps, $G$ is the window length, $d$ is the number of variables, and $L$ is the lag order. On the server side, since the server only aggregates the uploaded edge-level summaries across the $M$ clients and performs the corresponding global statistical tests for all candidate edges, the per-round complexity is $O\left(M(L+1)d^2\right)$. Therefore, the runtime is mainly dominated by client-side computation, while the server-side overhead remains relatively lightweight in federated settings. The per-round runtime on benchmark datasets reported below further supports this observation.
> |Dataset|Client time (s)|Serve rtime (s)|Total time (s)|
> |---|---:|---:|---:|
> |fMRI|325.83|0.16|325.99|
> |DREAM3|12969.15|0.59|12969.74|
>
> **Q5:** Can you illustrate if the assumptions you made in Appendix B are realistic in practice?
>
> **R5:** We thank the reviewer for this important question. The assumptions in Appendix B are intended as standard working assumptions that facilitate the theoretical analysis and modeling, rather than as claims that every real federated deployment satisfies them exactly. In particular, causal sufficiency is an idealized but commonly adopted assumption in causal discovery, and it has been used in many prior studies because it allows a clearer modeling framework and more tractable analysis. At the same time, it can still serve as a reasonable approximation in some practical federated settings where the studied subsystem is relatively well-instrumented and the main driving variables are routinely measured, such as multi-center clinical monitoring, industrial process monitoring, and smart-building systems. We also clearly recognize that unobserved confounding may arise in real federated environments, and extending our method to account for such settings is an important direction for future work.
>
> Similarly, the cross-client independence assumption under the null is introduced mainly to support the derivation of the null distribution of the meta-statistic. In the horizontal federated setting considered in our paper, where clients correspond to distinct data holders and raw data are not shared, this can be viewed as a reasonable approximation and does not require the client-specific dynamics themselves to be identical. In addition, the boundedness condition is a mild technical requirement used primarily for privacy calibration, and is generally consistent with finite-window Gaussian CI testing and regularized estimation in practice. We will clarify this in the revised manuscript.

---

> > ### Author Rebuttal · Reviewer_3kKz · 2026-04-02
> >
> > Thanks for the response. Most of my concerns are addressed. And I will keep my score.

---

> > > ### Author Response · Authors · 2026-04-05
> > >
> > > Thank you for recognizing our work and for your positive evaluation. We also sincerely appreciate your time and careful review of our paper.

---

### Decision · Program_Chairs · 2026-04-30

**Decision:**

Accept (regular)

**Comment:**

In this paper, the authors study multi-variate time series causal discovery in the federated learning setup that handles temporal drift and mismatched sampling rates across clients. Their approach combines local structure learning with credibility weighting across time lags, using matrix logarithm alignment to reconcile mismatched sampling and they weight edges using stability and lag consistency to learn the global causal graph without sharing raw data (under differential privacy). All the reviewers concerns were addressed in the rebuttal and the three reviewers converged on acceptance. The reviewers noted the strong theoretical assumptions and missing limitations sections.